# Nanowire-assisted electrochemical perforation of graphene oxide nanosheets for molecular separation

Hai Liu[1], Xinxi Huang[1], Yang Wang[1], Baian Kuang[1] & Wanbin Li[1] ✉

Two-dimensional nanosheets, e.g., graphene oxide (GO), have been widely used to fabricate efficient membranes for molecular separation. However, because of poor transport across nanosheets and high width-to-thickness ratio, the permeation pathway length and tortuosity of these membranes are extremely large, which limit their separation performance. Here we report a facile, scalable, and controllable nanowire electrochemical concept for perforating and modifying nanosheets to shorten permeation pathway and adjust transport property. It is found that confinement effects with locally enhanced charge density, electric field, and hydroxyl radical generation over nanowire tips on anode can be executed under low voltage, thereby inducing confined direct electron loss and indirect oxidation to reform configuration and composition of GO nanosheets. We demonstrate that the porous GO nanosheets with a lot of holes are suitable for assembling separation membranes with tuned accessibility, tortuosity, interlayer space, electronegativity, and hydrophilicity. For molecular separation, the prepared membranes exhibit quadruple water permeance and higher rejections for salts (>91%) and small molecules (>96%) as/than original ones. This nanowire electrochemical perforation concept offers a feasible strategy to reconstruct two-dimensional materials and tune their transport property for separation.

Molecular separations are vital for chemical, environmental, petrochemical, pharmaceutical, and energy-related industries[1–4]. Membrane technology, as a more efficient alternative process than energy-intensive processes, has been attracting increasing attention owning to its energy/cost-saving, small footprint, operational simplicity, etc. Polymeric membranes are widely used for molecular separations due to their superiorities in cost and processability, yet they undergo trade-off limitation between permeability and selectivity. Over the past decades, nanoporous materials, e.g., metal-/covalent organic frameworks[5–7], have been crystallized on substrates to form polycrystalline membranes for precise molecular sieving. However, the inferiorities in scalability, reproducibility, and controllability hinder their application.

Two-dimensional (2D) materials, e.g., graphene[8–10], graphene oxide (GO)[11–18], MXene[19], and molybdenum disulfide[20], can be used as building blocks to fabricate high-performance membranes. Their ultrasmall thickness, mechanical stability, and chemical tunability enable the membranes with great potential for liquid and gas treatments. Amongst, GO membranes are promising for efficient molecular separation, thanks to their appropriate interlayer space, nanosheet manufacturability, and membrane processability[2,11–18]. For GO membranes, the nanosheet stacking property determines that the transport pathways consist of interlayer channels and nanosheet defects/edges[21,22]. Considering the poor transport across nanosheets and high width-to-thickness ratio, the interlayer channels of GO membranes are dominant for separation. Molecules have to pass through the stenotic

[1]School of Environment, Jinan University, Guangzhou 511443, China. ✉e-mail: gandeylin@126.com

and circuitous paths between adjacent nanosheets to reach permeate sides. In other words, the tortuosity is extremely large. Various chemical and physical methods, e.g., reduction, oxidation, crosslinking, intercalation, and physical confinement, have been proposed to introduce pore structure and adjust interlayer space for enhancing permeability or selectivity/rejection[8–10,15–18,22–28]. Unlike nonporous 2D materials, some crystalline nanosheets, which are exfoliated from bulk crystals or crystallized from precursors, have intrinsic pores and allow fast molecular transports[29–31]. Unfortunately, the fabrication procedures are often complicated and the structural regularity is difficult to preserve. We envisage transformation of nonporous but easy-to-prepare nanosheets into porous ones in a facile route to boost separation performance.

Nanowire electrodes, made by metal oxides or metals with high aspect ratios, can simply execute lightning-rod effect under external voltages. Therefore, nanowire electrodes are regarded as practical platforms to perforate and destruct cells and target substances[32–37], e.g., bacterial electroporation[32,33]. Different from conventional electroporation with plate electrodes, which needs working voltage over 10 kV, nanowires can perform confinement effects over tips, through driving the migration of free electrons by low external voltage (< 10 V), to induce the formation of gravitated charge density and enhanced electric field[38]. Meanwhile, the concentrated hydroxyl radicals (•OH) can be generated by the $H_2O$ and $OH^-$ electrochemical reactions in the finite regions at nanowire tips to promote the local oxidation of targeted matters[34,35]. Moreover, electric attraction can enrich charged target substances on nanowire electrodes with oppositely charged property to amplify perforation. In this study, we report nanowire electrochemical perforation concept to transform nonporous GO nanosheets to porous ones in a facile and controllable route (Fig. 1 and Supplementary Fig. 1). It is found that nanowire electrochemical perforation not only causes the formation of holes and defects in nanosheets but also tunes their chemical composition and topological configuration, thereby improving the permeance and rejection of prepared membranes.

## Results

### Nanowire perforation of GO nanosheets

Nanowire perforation of GO nanosheets was conducted by using an electrochemical apparatus with titanium (Ti) net cathode and tricobalt tetraoxide nanowire ($Co_3O_4$-NW) anode (Fig. 2a–c, and Supplementary Fig. 2). Porous graphite felts with fibre diameters of ~10 μm and pore sizes of 50–200 μm were used as substrates for supporting $Co_3O_4$-NW (Fig. 2a and Supplementary Fig. 2), because their high surface area was beneficial to the $Co_3O_4$-NW deposition and their large pores allowed the GO nanosheets to pass through. To fabricate nanowire anode, $Co_3O_4$-NW was vertically grown on the felt fibres by hydrothermal treatment and calcination. X-ray diffraction (XRD) and scanning electron microscopy (SEM) characterizations corroborated that the nanowires were of well crystallinity and had diameter of ~40 nm, length of ~3.3 μm, and density of ~3 wires μm$^{-2}$ (Supplementary Fig. 2,3). For nanowire perforation, the GO suspension with concentration of 20 μg mL$^{-1}$ sequentially flowed through the pores of cathode and anode (Fig. 2a). With applying voltages, the free electron at cathode would promote the generation of $OH^-$ via $H_2O$ electrolysis. Then, the formed $OH^-$ and negatively charged GO nanosheets flowed to anode, and were prone to migrate toward the nanowire tips under electrostatic attraction and electrophoretic force. Meanwhile, the presence of $OH^-$ would further improve the electronegativity of GO. Consequently, the nanowires could conduct lightning-rod effect to perforate the accumulated GO nanosheets, through direct electrochemical reaction from accelerating electron loss. As well, the nanowire anode could capture electrons from $OH^-$ to generate •OH for implementing indirect oxidation of GO nanosheets. We measured the pH of the GO suspensions at influent, cathode-treated influent, and effluent according to in-situ sampling experiments (Supplementary Fig. 2). Obviously, the pH increased from 4.5 for influent to 10–11 for cathode-treated influent at applied voltages of 3–9 V and then decreased to ~4.5 for effluent (Fig. 2d). This pH alteration revealed the generation of a large amount of $OH^-$ from $H_2O$ electrolysis at cathode and the $OH^-$ consumption to form $O_2$ or •OH at nanowire anode.

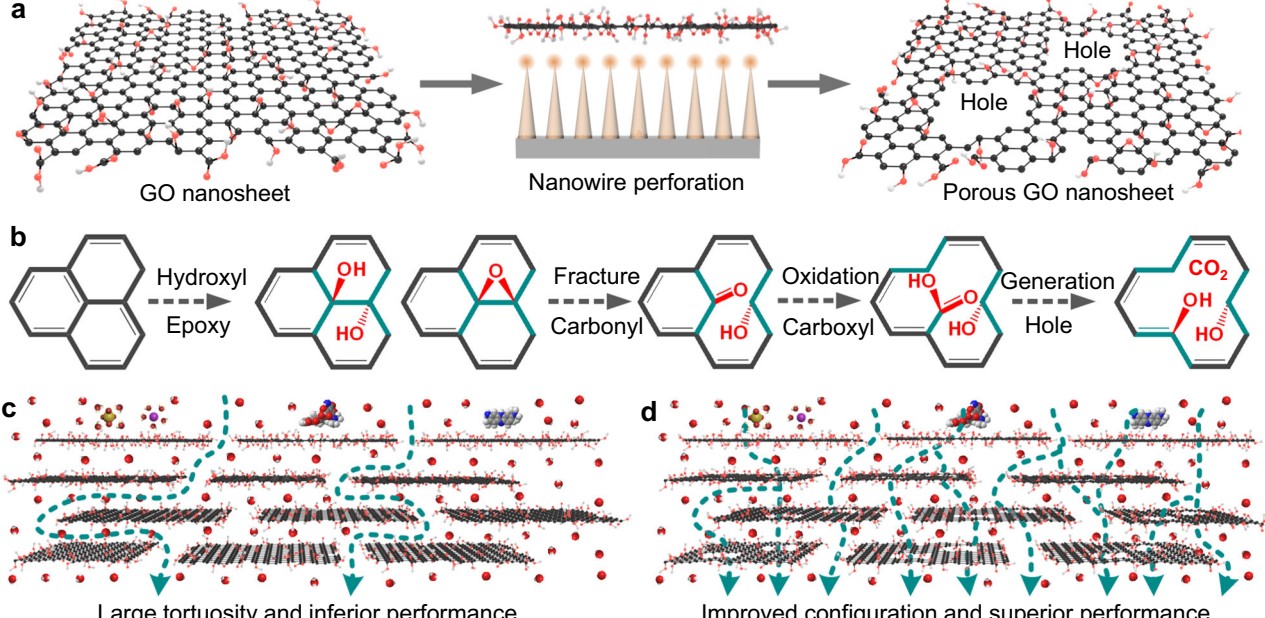

**Fig. 1 | Nanowire perforation of GO nanosheets and transport mechanism of related membranes. a** Schematic of nanowire electrochemical perforation for transforming nonporous GO nanosheet to porous one. **b** Schematic of graphene oxidation process, including hydroxyl and epoxy dosing, carbonyl and carboxyl formation from carbon-carbon fracture, nanopore generation from decarboxylation. **c**, **d** Schematics of molecular transport through (**c**) conventional GO and (**d**) porous GO membranes. Black balls, red balls, and dark cyan arrows represent carbon atoms, oxygen atoms, and water transport processes.

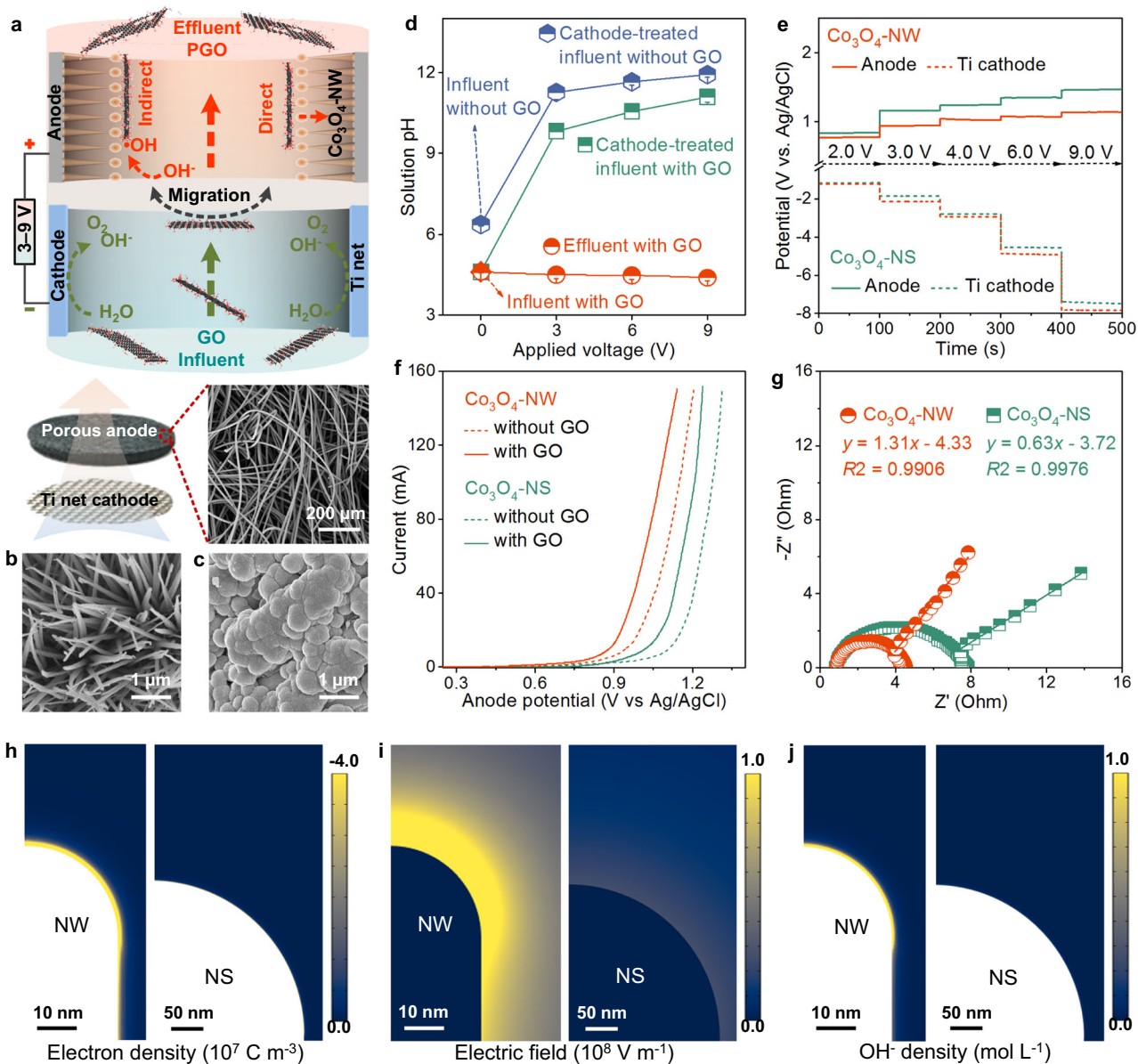

**Fig. 2 | Electrochemical characterizations and finite element simulations.**
**a** Schematic illustration of nanowire perforation for GO nanosheets by using an electrochemical apparatus with a flow-through mode. GO nanosheets in suspension flowed through the cathode and anode sequentially, and were transformed to porous ones at $Co_3O_4$-NW anode with applied voltages of 3–9 V. **b**, **c** SEM images of $Co_3O_4$-NW and $Co_3O_4$-NS on fibre of porous graphite felt (SEM in (**a**)). **d** Solution pH of influent, influent after treatment with Ti net cathode, and effluent after treatment with $Co_3O_4$-NW anode. Error bars are standard deviations from three tests. **e** Potential distributions of the electrodes under voltages from 2.0 to 9.0 V. **f**, **g** Linear scanning voltammetry (LSV) curves and electrochemical impedance spectra of the $Co_3O_4$-NW and $Co_3O_4$-NS anodes. **h–j** Distributions of finite (**h**) electron density (**i**) electric field, and (**j**) $OH^-$ density on $Co_3O_4$-NW and $Co_3O_4$-NS anodes. NW and NS in (**b–j**) represent nanowires and nanospheres, respectively.

Various electrochemical analyses and theoretical simulations were carried out to demonstrate the finite-concentrated electrochemical reactions over nanowire tips. To illustrate the confinement effects of nanowires, the $Co_3O_4$ nanospheres ($Co_3O_4$-NS) with diameter of 400–450 nm were deposited on graphite felt and used as anode for comparison (Fig. 2c). Distributions of electrode potential and current are related to reaction potential and strength. Under applied voltages of 2–9 V, $Co_3O_4$-NW had lower anodic potential and higher current than $Co_3O_4$-NS (Fig. 2e and Supplementary Fig. 4), reflecting the promotion of nanowires to electrochemical reactions. As mentioned above, nanowire perforation of GO nanosheets was conducted through two pathways of direct and indirect •OH oxidations. Linear scanning voltammetry (LSV) can explore polarization potential, which means the occurrence of reactions. Theoretically, the acidic

substances of GO nanosheets would neutralize the $OH^-$ of cathode-treated influent and then increase the electrode potential for $OH^-$ oxidation. As expected, the experimental results confirmed that the GO addition reduced the pH and the $OH^-$ concentration of cathode-treated influent (Fig. 2d). However, the polarization potential (-0.91 V vs. Ag/AgCl) of $Co_3O_4$-NW for the GO suspension was lower than that for the solution without GO ( ̃ 1.01 V vs. Ag/AgCl). This phenomenon validated the existence of direct GO electrochemical reaction (Fig. 2f). For the solution without GO, the $Co_3O_4$-NW anode displayed lower potential (-1.01 V vs. Ag/AgCl) for $OH^-$ electrolysis occurrence than $Co_3O_4$-NS ( ̃ 1.15 V vs. Ag/AgCl) as well. In fact, no matter whether there was GO in solution or not, the potentials of $Co_3O_4$-NW were lower than that of $Co_3O_4$-NS, which attested to the confinement effects over nanowire tips. Electrochemical impedance spectra with an incomplete

semicircle and a straight line for their Nyquist plots were associated with charge transfer resistance and diffusion resistance for GO and OH$^-$ oxidations. A smaller semicircle (radius of ~1.9 Ω) and a more vertical line (slope of 1.31) of the Co$_3$O$_4$-NW anode than those of Co$_3$O$_4$-NS (~4.1 Ω and 0.63) suggested its faster charge transfer and diffusion for anodic reactions (Fig. 2g).

We studied the finite-concentrated electrochemical reactions on nanowire tips by finite element simulations of tip-enhanced charge density, field intensification, and OH$^-$ density (Fig. 2h–j)[36,37]. The Co$_3$O$_4$-NW anode possessed the locally charge density, electric field, and OH$^-$ density up to $10^7$ C m$^{-3}$, $10^8$ V m$^{-1}$, and 1 mol L$^{-1}$, respectively, which were two to three orders of magnitude as those of Co$_3$O$_4$-NS. Such high finite-concentrated reaction activity over nanowire tips could significantly facilitate the nanoscale perforation for negatively charged GO via direct and indirect reactions[39,40]. All above electrochemical analyses and simulations verified that the finite-concentrated reactions over nanowire tips could induce the reaction of locally enriched GO nanosheets on nanowire anode.

## Configuration and composition of porous GO nanosheets

We perforated GO nanosheets by nanowire electrodes at applied voltages of 3–9 V ($x$) to obtain porous GO-$x$ (PGO-$x$) nanosheets. Atomic force microscope (AFM) was employed to detect the configuration. Obviously, the GO nanosheets were monolayer with thickness of ~1.0 nm (Fig. 3a and Supplementary Fig. 5). After perforation, the PGO nanosheets became thicker (~2.0 nm) (Fig. 3a–h, and Supplementary Fig. 5), originating from the grafted out-of-plane oxygen-containing groups. It should be noted that many uniform holes were generated in the PGO-6 and PGO-9 nanosheets and became larger as voltage raised, with average diameters of 35 and 85 nm, respectively (Supplementary Fig. 6). Considering the 37.4% area ratio of holes from ImageJ, the density of holes in PGO-6 could be approximately estimated as $4.0 \times 10^{14}$ m$^{-2}$. These holes would be helpful for permeability. Low voltage of 3 V was insufficient to produce AFM-detectable holes. Unlike PGO-9, the depth of holes in PGO-6 was smaller than the thickness (Fig. 3g), owing to the inaccessibility of holes toward AFM tips. We observed the GO and PGO nanosheets by transmission electron microscope (TEM) (Fig. 3i–l, and Supplementary Fig. 7). It has been proved that GO nanosheets have both graphitic domains and epoxy/hydroxyl patches, and the carbonyl and carboxyl groups often locate at GO defects and edges[41–43]. As displayed in the TEM images (Fig. 3i), the GO nanosheets possessed both compact and loose skeletons, assigned to the graphitic and oxygen-containing domains, respectively. Relatively, the compact regions of PGO-3 shrunk (Fig. 3j and Supplementary Fig. 7), implying the electrochemical oxidation. More white speckles in the PGO-3 nanosheets were ascribed to the more defects from the fracture of carbon-carbon bonds. For the PGO-6 and PGO-9 nanosheets, the holes could be observed obviously (Fig. 3k,l). It was noteworthy that the TEM-detected holes were narrower than those from AFM, resulted from the effects of the Van Der Waals interaction and the curvature radius of tips over 10 nm. Nanowire electrochemical perforation of GO depended on reaction strength and reaction time. Besides perforation at various applied voltages for GO suspension with flow rate of 15 mL min$^{-1}$ and retention time of 9.8 s, nanowire perforation was further performed with flow rate of 30 mL min$^{-1}$ and retention time of 4.9 s. As expected, the PGO-6 nanosheets with shorter retention time showed smaller holes (Supplementary Fig. 8). These results demonstrated the controllability of nanowire electrochemical perforation.

X-ray photoelectron spectroscopy (XPS) was used to characterize the chemical binding states. After electrochemical perforation, the PGO nanosheets had higher oxidation degree than GO, especially for PGO-3 (Supplementary Table 1). By peak-differentiating and imitating, the C 1s XPS spectra could be fitted by typical peaks of C-H/C-C/C = C (284.8 eV), C-OH/C-O-C (286.7 eV), and $C$ = O/COOH (287.6 eV)

(Fig. 3m); and the O 1s profiles could be fitted by three peaks of C-O/C-O-C, $C$ = O, and COOH (Supplementary Fig. 9). In accordance with the oxidation degree, the peak areas of oxygen-containing groups in the C 1s profiles of PGO were expanded, and PGO-3 had highest C-OH/C-O-C/C = O/COOH content of 51.2% (Fig. 3m,n). For the O 1s spectra, the $C$ = O and COOH peaks of PGO were more intensive than those of GO, and the variations were similar to those for C 1s (Fig. 3o and Supplementary Fig. 9). The similar chlorine content, the nonexistence of C-Cl in C 1s XPS, and the similar Cl $2p$ peak shape implied the limited reaction between Cl$^-$ and GO (Supplementary Fig. 10). The higher sodium content than chlorine one was explained by the stronger interaction of GO to Na$^+$ (Supplementary Fig. 11). As reported in previous study[44], the pinhole defects that were not AFM-detectable could be roughly estimated by using the edge carbon atoms with double bonded oxygen. So the area ratio of pinhole defects in GO, PGO-3, PGO-6, and PGO-9 were approximately 8.3%, 16.0%, 11.8%, and 12.8%, respectively.

Graphene oxidation process can be roughly divided into three steps (Fig. 1b): i, hydroxyl and epoxy decoration, ii, carbon-carbon fracture and further oxidation to form carbonyl and carboxyl, and iii, decarboxylation to generate nanoholes[45–47]. Comprehensively considering the results of AFM, TEM and XPS, we could deduce the reaction mechanism as below (Fig. 3p–s). For nanowire perforation with voltage of 3 V, i and ii reactions were dominant, thereby leading to the enrichment of oxygen-containing groups and pinhole defects without formation of detectable holes (Fig. 3q). When the voltage was 6 V or 9 V, stronger direct and indirect oxidations over nanowire tips could promote the simultaneous occurrence of i, ii, and iii reactions, thus causing hole generation (Fig. 3r,s). Because strong electrochemical reactions over nanowire tips led to oxidation and perforation for both graphitic and oxygen-containing regions, the oxygen-containing groups of PGO-6 and PGO-9 varied slightly.

So far, some other methods, e.g., chemical etching and air oxidation, have also been reported for generation of holes in GO nanosheets (Supplementary Table 2)[48–53]. Compared with hydrogen peroxide-needed chemical etching and high-temperature air oxidation, which had intermittent batch processing for several hours, the nanowire electrochemical strategy reported here, with continuous flow-through procedures, room temperature condition, and GO suspension flux over 1800 L m$^{-2}$ h, might have superiorities in operability, controllability, scalability, environmental friendliness, economy, safety, and efficiency. Moreover, relative to the previously reported methods, which might involve the reactions at oxygen-containing regions of GO or at defect regions of graphene, the nanowire electrochemical method could perform oxidation and perforation for both graphitic and oxygen-containing regions because of the strong electrochemical reactions over nanowire tips.

## Assembly and property of PGO membranes

For assembly of membranes, the nanosheets were deposited on polyethersulfone microfiltration membranes with 0.22-μm pores through vacuum filtration. Photographs presented the uniform coverage of GO and PGO nanosheets on substrate surfaces (Supplementary Fig. 12). As illuminated in the SEM images (Fig. 4a–d, and Supplementary Fig. 13–15), all GO and PGO membranes with different loadings were ultrathin and defect-free. Some sunken imprints and winkles were observed on membranes (Supplementary Fig. 13–15), which were derived from the vacuum suction through the pores of substrates and the evaporation of water between flexible GO nanosheets. Relative to the GO membrane, the PGO membranes were thicker (Fig. 4a–d), which agreed with the nanosheet configuration from AFM.

Interlayer space is main transport pathway for GO membranes. The XRD patterns indicated that the interlayer space of the GO membranes was 0.91 nm, from the peak at 9.7° (Fig. 4e and Supplementary Fig. 16,17). After electrochemical perforation, the peak shifted

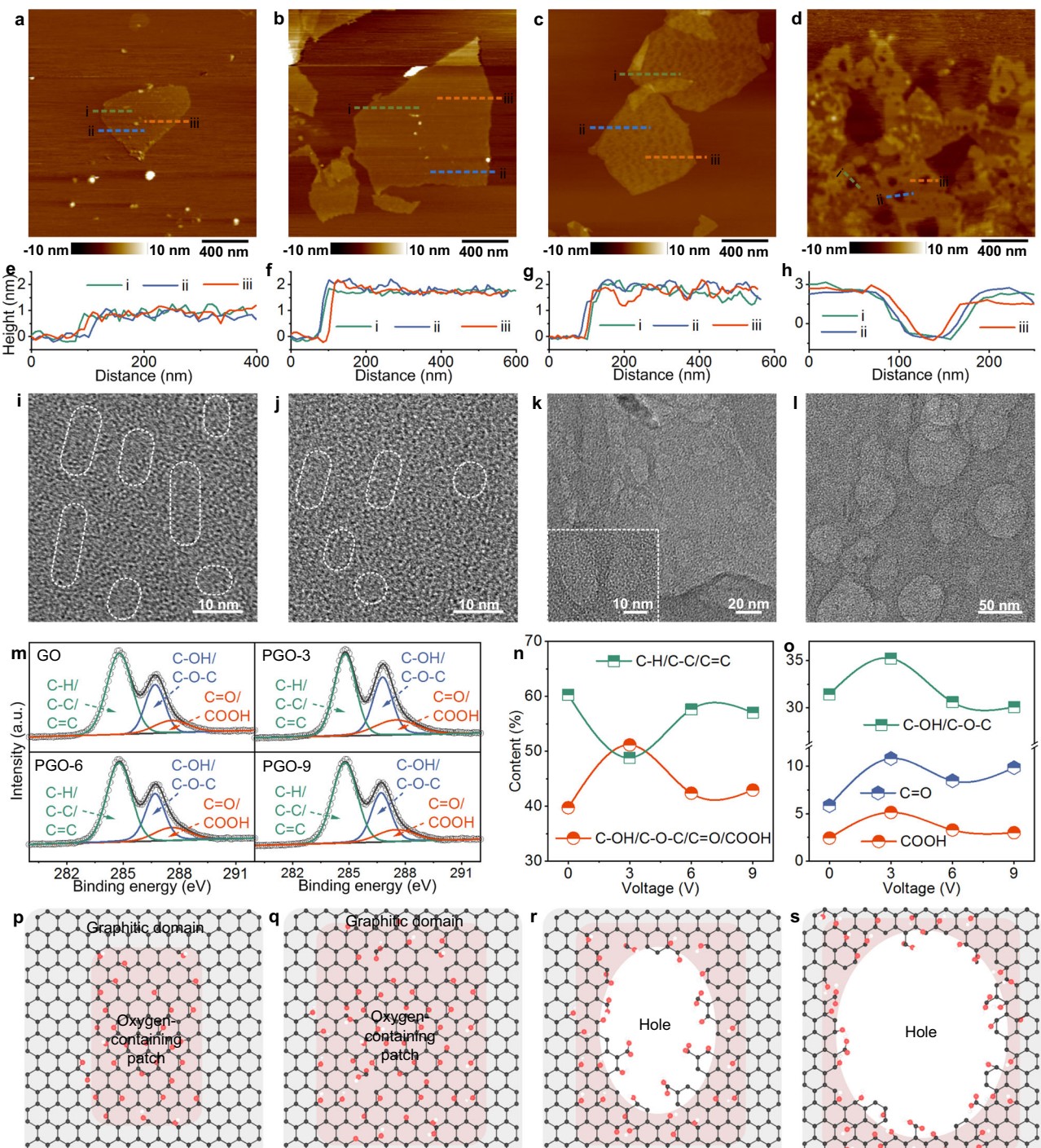

**Fig. 3 | Characterizations of GO and PGO nanosheets. a–d** AFM images of the GO, PGO-3, PGO-6, and PGO-9 nanosheets. Green, blue, and red dashed lines are marked by i, ii, and iii, respectively. **e–h** Height profiles of the GO, PGO-3, PGO-6, and PGO-9 nanosheets. i, ii, and iii represent three positions that are marked in the AFM images. **i–l** TEM images of the GO, PGO-3, PGO-6, and PGO-9 nanosheets. **m** C 1 s XPS spectra of the GO, PGO-3, PGO-6, and PGO-9 nanosheets. a. u. represents arbitrary unit. **n** C-H/C-C/C = C and C-OH/C-O-C/C = O/COOH contents of the GO and PGO nanosheets from C 1 s XPS. **o** C-OH/C-O-C, C = O, and COOH contents of the GO and PGO nanosheets from O 1 s XPS. **p–s** Schematic illustration of chemical structures and configurations of the (**p**) GO (**q**) PGO-3 (**r**) PGO-6, and (**s**) PGO-9 nanosheets. Graphitic domains, epoxy/hydroxyl patches, and holes of GO and PGO are marked by dark grey, light red, and white, respectively.

to 9.2° for PGO-3 and PGO-6, suggesting the expanded interlayer space of 0.96 nm. Notably, the characteristic peak for the PGO-9 membranes almost disappeared, as the large holes and dilapidated configuration induced excessively irregular nanosheet arrangement. In terms of the filtration condition, the interlayer space of the membranes after water swelling was investigated. As shown in the XRD patterns, the hydration of nanosheets caused interlayer space expansion. The wetted GO,

PGO-3, and PGO-6 membranes had interlayer space of 1.28 (6.9°), 1.32 (6.7°), and 1.38 (6.4°) nm, respectively. Raman spectroscopy can assess GO structure regularity. Two typical peaks for *D* and *G* bands were obviously observed for all GO and PGO membranes (Fig. 4f). These *D* and *G* bands from the breathing vibration and in-plane stretching vibration of carbon were associated with the defect/disordered $sp^3$ and original $sp^2$ networks, respectively[54]. In consistence with the XPS

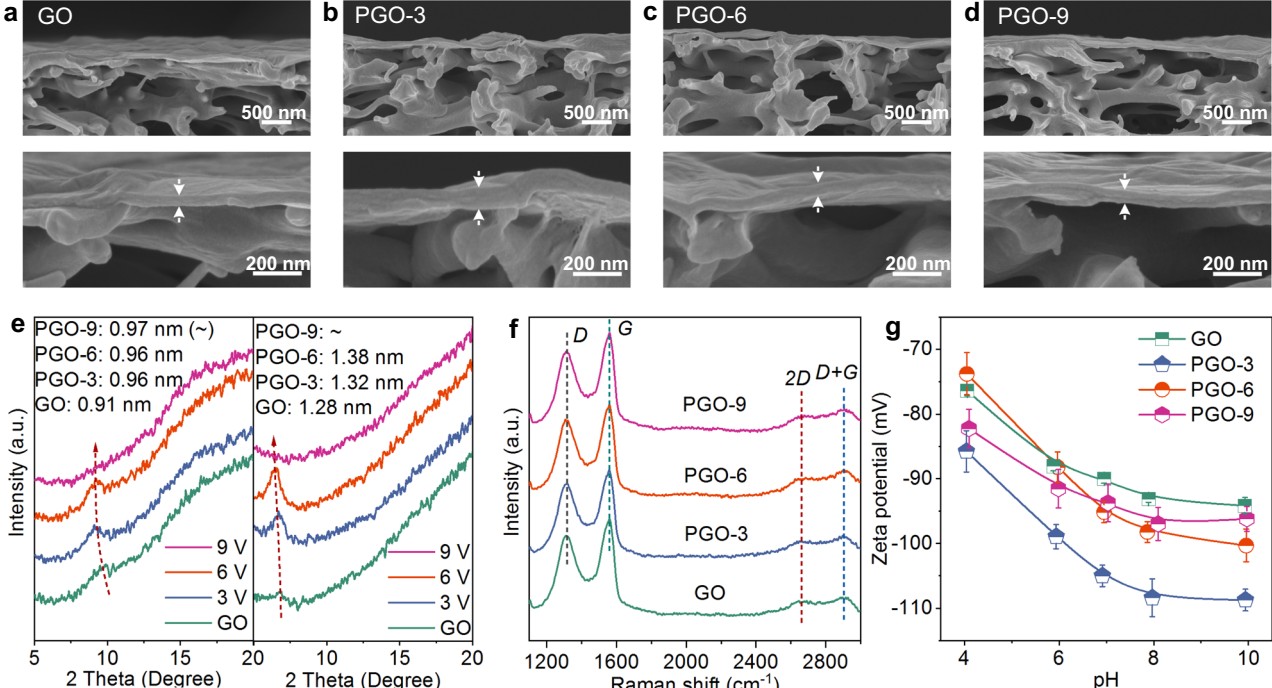

**Fig. 4 | Characterizations of GO and PGO membranes. a–d** Cross-sectional SEM images of the GO, PGO-3, PGO-6, and PGO-9 membranes. All membranes were prepared with loading of 200 µg. **e** XRD patterns of the GO, PGO-3, PGO-6, and PGO-9 membranes at dry and wetted states. Interlayer space values of different membranes are presented. **f** Raman spectra of the GO and PGO membranes. *D, G, 2D*, and *D + G* bands are marked. **g** Zeta potential of the GO and PGO membranes under different pH. Error bars are standard deviations calculated from three samples. a. u. in (**e**, **f**, **g**) represents arbitrary unit.

results, the PGO-3 membrane had maximum $I_D/I_G$ intensity ratio and most disorder structure due to the most functional groups (Supplementary Table 3). Two peaks of *2D* and *D + G* bands, that correlated out-of-plane stacking regularity (*2D*) and irregularity (*D + G*) of graphitic domains, occurred in Raman spectra of all membranes[55]. As more generation of holes caused poorer stacking arrangement, the $I_{2D}/I_{D+G}$ ratio of the PGO membranes became attenuated as voltage upgraded (Supplementary Table 3).

For membrane separation, surface charge property affects performance through electrostatic repulsion (known as Donnan effect). The membranes with massive oxygen-containing groups were negatively charged at pH from 4 to 10 (Fig. 4g). Consistent with chemical structures, the PGO-3 membrane with most oxygen-containing groups had maximum electronegativity among four membranes. Surface hydrophilicity is important for membrane separation. All membranes were hydrophilic (contact angle, *CA* < 90°) and the PGO membranes were more hydrophilic than GO (Supplementary Fig. 18). On account of the oxygen-containing group content, the hydrophilicity of the PGO-3 membrane should be better than other membranes. In fact, dynamic water contact angle measurement showed that the PGO-6 membrane had smallest CA and fastest CA declining rate as time extended. This phenomenon could be attributed to that the hydrophilic holes in the PGO-6 nanosheets facilitated the water transport through the PGO-6 layer into the porous substrate. An analogous result has been observed for other porous materials[25].

### Separation performance of PGO membranes

We evaluated the performance of the GO and PGO membranes for Na₂SO₄ desalination (Fig. 5a–c, and Supplementary Fig. 19). A cross-flow nanofiltration apparatus was operated with cross-flow velocity of ~1.2 cm s⁻¹, which was two orders of magnitude as that for practical filtration (0.01 cm s⁻¹)[24]. The GO membrane with loading of 200 µg (14.4 µg cm⁻²) had permeance of 8.0 L m⁻² h⁻¹ bar⁻¹ and rejection of 67.9%. Interestingly, the PGO membranes showed simultaneously

ameliorated permeance and rejection. For examples, the PGO-3 and PGO-6 membranes displayed double and quadruple permeance of 15.5 and 34.6 L m⁻² h⁻¹ bar⁻¹ and high rejection of 96.5% and 92.9%, respectively. Comparatively, the performance of PGO-9 was relatively poorer due to the excessive irregular arrangement. When the loading amount was 100 µg (7.2 µg cm⁻²), the permeance and rejection of the GO membrane were 12.8 L m⁻² h⁻¹ bar⁻¹ and 65.9%, respectively; while the PGO-6 membrane still exhibited quadruple permeance of 49.4 L m⁻² h⁻¹ bar⁻¹ and rejection of 91.6%. After further reducing the loading amount to 50 µg (3.6 µg cm⁻²), the permeance of the PGO-6 membrane reached up to 70.1 L m⁻² h⁻¹ bar⁻¹, despite the deterioration of rejection to 73.1%. As feed pressure increased, the water flux of the GO and PGO membranes grew linearly, revealing the pressure stability (Fig. 5d and Supplementary Fig. 20–23); but the rejection and permeance decreased slightly, in virtue of the more compact selective layers and the more severe concentration polarization phenomenon[16,56,57]. Because of the existence of holes, the PGO membranes had slower permeance declining rate as pressure raised (Supplementary Fig. 23). Figure 5e indicated the long-term application prospect of the PGO membranes. After nanofiltration for 80 h, the rejection and permeance still kept at 93.9% and 40.2 L m⁻² h⁻¹ bar⁻¹, respectively. Several times of permeance increment and decrement during filtration with non-consecutive operation mode were the results of the loosening of nanosheets in solution with no pressure and the tightening under pressure, respectively, which were consistent with the permeance variation at the beginning of filtration (Supplementary Fig. 19).

To transport through whole GO membranes, molecules and ions have to pass through vertical defects/edges and shuttle back and forth in horizontal interlayer channels between adjacent nanosheets[21,22]. Compared with the GO nanosheets, the holes and defects of PGO could serve as interconnected channels to reduce the transport distance and membrane tortuosity for improving permeance. For GO and PGO membranes, the transport pathway length (*S*) could be calculated based on membrane thickness (*h*), interlayer space (*d*), nanosheet size

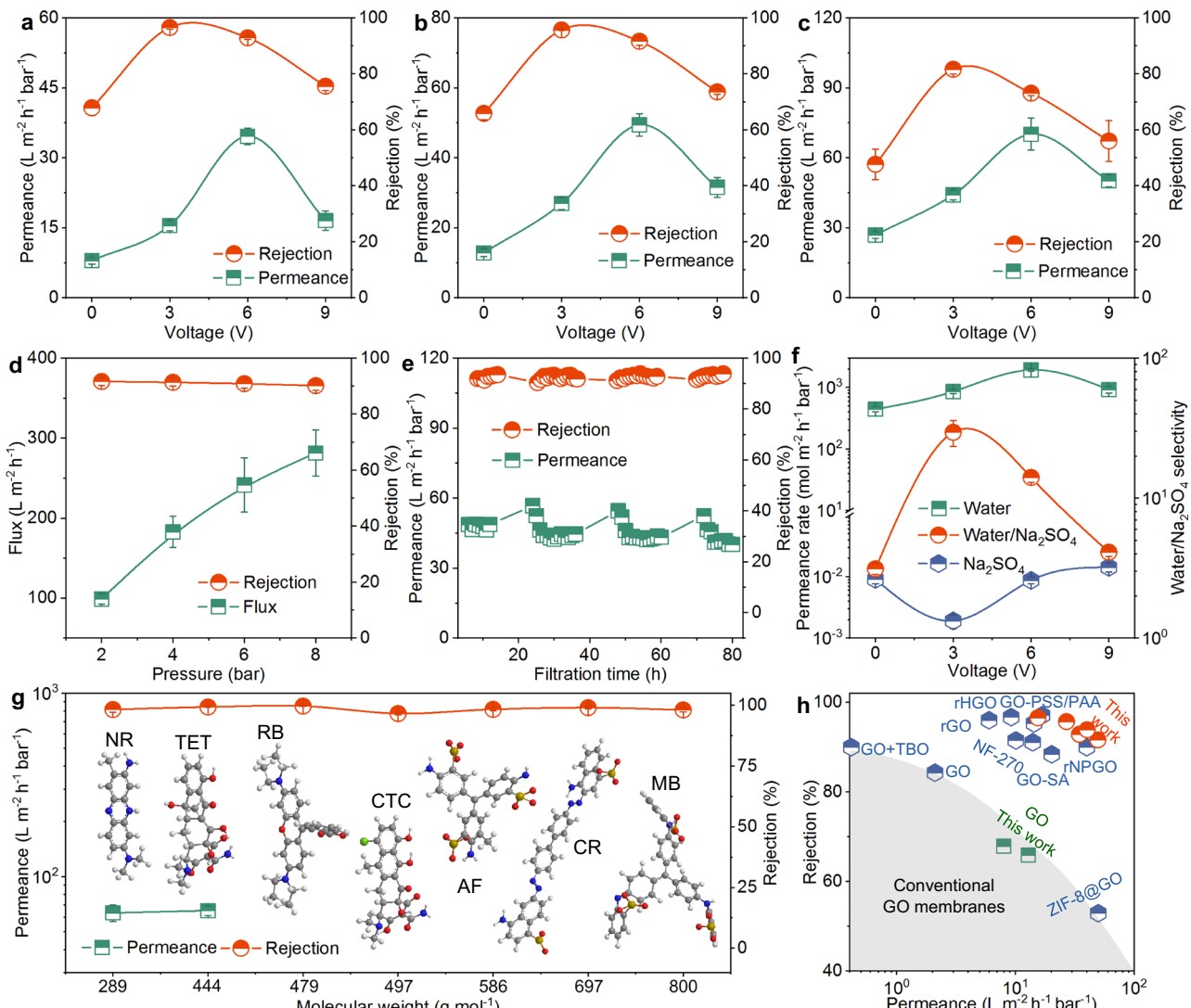

**Fig. 5 | Separation performance of GO and PGO membranes. a–c** Water permeance and Na$_2$SO$_4$ rejection of the GO and PGO membranes prepared with loadings of (**a**) 200 (**b**) 100, and (**c**) 50 μg at room temperature and feed pressure of 2 bar. **d** Water flux and Na$_2$SO$_4$ rejection of the PGO-6 membrane prepared with loading of 100 μg at different feed pressures. **e** Long-term filtration of the PGO-6 membrane prepared with loading of 100 μg for Na$_2$SO$_4$ solution at pressure of 2 bar. Permeance increments were attributed to the loosening of nanosheets in solution with no pressure. **f** Water permeation. rate, Na$_2$SO$_4$ permeation rate, and water/Na$_2$SO$_4$ selectivity of the GO and PGO membranes prepared with loading of 200 μg at feed pressure of 2 bar. **g** Separation performance of the PGO-6 membrane prepared with loading of 50 μg for antibiotic remediation and dye removal at pressure of 2 bar. NR, TET, RB, CTC, AF, CR, and MB are neutral red, tetracycline, rhodamine B, chlortetracycline hydrochloride, acid fuchsin, Congo red, and methyl blue, respectively. **h** Comparison of the PGO membranes with commercial, GO, and porous GO membranes for Na$_2$SO$_4$ desalination. Desalination performance of most GO membranes is located in the dark grey region. Some recently reported GO membranes after modification with advanced separation performance are presented specially (Supplementary Table 4). TBO, SA, PAA, PSS, rHGO, and rNPGO are toluidine blue O, sodium alginate, polyacrylic acid, polystyrene sulfonate, reduced holey GO membranes, and reduced nanoporous GO membrane, respectively. All average performance data in (**a**–**f**, **g**) with error bars of standard deviations were calculated from three membrane samples.

($L$, assumed as 1 μm), and hole density ($\rho$) as equations of $S = h/d \times L$ and $S = h/d \times L/((L^2 \times \rho)^{0.5} + 1)$, respectively[49,51]. It was calculated that the transport pathway length of PGO-6 was only about 5% as that of GO. In other words, the tortuosity of the PGO-6 membrane was one twentieth as that of GO. Relative to the reduction in pathway length, the quadruple permeance increment of PGO-6 as GO was lower. This might be resulted from that the pinhole defects in the GO and PGO nanosheets and the free spaces from a certain irregular stacking could serve as channels to increase the water permeance of the GO and PGO membranes[58]. Moreover, the expanded interlayer channels would also be beneficial to water permeability. Less permeance upgradation of PGO-3 than PGO-6 was explained by their inferior porosity; while the poorer permeance of PGO-9 than PGO-6 was attributed to the excessive membrane irregularity from nanosheet damage and the

occupation of neighbouring nanosheets to large space of holes. For Na$_2$SO$_4$ desalination, separation is usually governed by size exclusion and Donnan effect. For GO membranes, size exclusion might be not a dominated mechanism for rejection[21], especially considering the peeling of water from the shell of hydrated ions. We measured the NaCl desalination performance and investigated the effects of Na$_2$SO$_4$ concentration and solution pH on performance of the PGO-6 membrane. Low NaCl rejection of 35.2% and decreased rejection for Na$_2$SO$_4$ solution with higher concentration and lower pH suggested that the nanofiltration was mainly affected by Donnan effect[51] (Supplementary Fig. 24,25). Strong electrostatic repulsion of negatively charged surface towards SO$_4^{2-}$ endowed the membranes with good rejection. Consequently, the maximum rejection was achieved by the PGO-3 membrane with largest electronegativity. It should be pointed out that

the slightly smaller zeta potential of PGO-6 than GO at pH of ~7 seemed difficult to bring about substantial rejection amelioration. We calculated the permeation rates and water/salt selectivity of various membranes with different thickness under various feed pressures (Fig. 5f and Supplementary Fig. 26). Corresponding to permeance and rejection, the water permeation rate and water/salt selectivity had analogous changing trend with the membrane thickness and perforation voltage. For instance, the selectivity of the PGO-3 membrane was an order of magnitude as that of the GO membrane, up to 29.5. Differently, the $Na_2SO_4$ permeation rate of various membranes was roughly ordered by PGO-3 < GO < PGO-6 < PGO-9. A similar or lower $Na_2SO_4$ permeation rate of PGO-6 than PGO-9 was attributed to that the more regular stacking provided more uniform electronegative sub-nanometre interlayer channels for better electrostatic repulsion. Because the increment of water permeation rate was greater than that of $Na_2SO_4$, PGO-6 had better rejection and selectivity. In other words, the water permeation enhancement contributed to the rejection and selectivity of the PGO-6 membrane.

Apart from desalination, we further investigated the performance of the prepared membranes for other separation applications, including water remediation for antibiotics and wastewater purification for dyes (Fig. 5g and Supplementary Fig. 27). Tetracycline and neutral red with molecular weights (MW) of 444 and 288 g mol$^{-1}$ were selected as main pollutants. For tetracycline remediation, the PGO-6 membrane with loading of 50 μg exhibited competitive rejection of 99.3% and permeance of 65.3 L m$^{-2}$ h$^{-1}$ bar$^{-1}$, respectively. The slight permeance deterioration relative to that for $Na_2SO_4$ desalination was interpreted by the sedimentation of antibiotic molecules on membrane surface during filtration. For neutral red removal, the permeance and rejection of the PGO-6 membrane were achieved at 63.4 L m$^{-2}$ h$^{-1}$ bar$^{-1}$ and 98.3% and were substantially superior to those of the GO membrane. It should be noted that the dye solution concentration became slightly higher after filtration (55–60 mg L$^{-1}$ varied for different experiments) due to the loss and sample collection of permeate solution, suggesting the separation was mainly based on rejection. As reported in previous studies[44,59], the adsorption of membranes for antibiotics and dyes occurred during filtration. Certainly, this adsorption could change the structure, interlayer space, and chemical composition of membranes, thereby affecting and contributing to separation. Besides these two organic molecules, the PGO-6 membrane could efficiently remove other antibiotics and dyes, with rejections of 99.7%, 96.6%, 98.3%, 99.0%, and 98.1% for rhodamine B (MW: 479 g mol$^{-1}$), chlortetracycline hydrochloride (MW: 497 g mol$^{-1}$), acid fuchsin (MW: 586 g mol$^{-1}$), Congo red (MW: 697 g mol$^{-1}$), and methyl blue (MW: 800 g mol$^{-1}$), respectively. It was worth emphasizing that the separation efficiency of the membranes was well for both negatively and positively charged molecules and for tetracycline solutions with different pH, confirming that the size exclusion was critical to antibiotic and dye removals. Under cross-flow filtration, the GO membranes often had good rejection over 90% for large-sized dyes (e.g., methyl blue), yet the rejection for salts or small molecules (MW < 350 g mol$^{-1}$) was usually inferior. Our PGO membranes showed superior desalination and nanofiltration performance, especially for $Na_2SO_4$ and small molecules. Moreover, compared with recently reported high-performance GO-based membranes after chemical modification and hole generation (Fig. 5h and Supplementary Table 4), the separation efficiency of the PGO membranes were still exceptional and competitive.

## Discussion

We have developed a simple nanowire perforation concept for transformation of nonporous GO nanosheets to porous ones. Nanowire electrodes can form strong confinement effects with locally enhanced charge density, electric field, and OH$^-$ density over nanowire tips, thereby generating nanoholes and defects in the enriched GO

nanosheets on nanowire anode and tuning their chemical structure and configuration, through direct reaction from losing electrons and indirect oxidation from •OH radicals. Because the formed holes and grafted functional groups can provide interconnected channels to shorten transport distance and membrane tortuosity, expand interlayer space, boost electronegativity and hydrophilicity, and adjust arrangement regularity, the PGO membranes exhibit substantially superior separation efficiency for various applications, including desalination, water remediation, and wastewater purification. For example, the PGO-6 membrane has quadruple water permeance as original GO membrane and exceptional rejections, up to 96.5 % for $Na_2SO_4$, 99.3% for antibiotic, and 99.7% for dye. Overall, the nanowire electrochemical perforation concept offers a scalable and controllable route to construct nanoholes in 2D materials and tune their chemical structures for improving their separation performance and other properties.

## Methods
### Materials
Cobalt nitrate hexahydrate [$Co(NO_3)_2 \cdot 6H_2O$], ammonium fluoride ($NH_4F$) and urea [$CO(NH_2)_2$] were purchased from Macklin Biochemical Reagent Co., China. Graphite felts were purchased from Jingu Carbon Material Reagent Co., China, as porous substrates for synthesis of nanowire-modified electrodes. Sodium chloride (NaCl, ≥99.5%) and sodium sulfate ($Na_2SO_4$, ≥99.0%) were purchased from Guanghua Chemical Reagent Co., China. GO powder was purchased from XFNANO Material Reagent Co., China. Polyethersulfone membrane was purchased from Kutai Chemical Reagent Co., China. The above chemicals used were of analytical grade unless otherwise stated. All aqueous solutions were prepared with deionized (DI) water (resistance ≥ 18.2 MΩ cm$^{-1}$).

### Fabrication of $Co_3O_4$-NW and $Co_3O_4$-NS electrodes
$Co_3O_4$-NW or $Co_3O_4$-NS was deposited on the porous graphite felt through a two-step method. For $Co_3O_4$-NW deposition, the precursor solution was prepared by adding $Co(NO_3)_2$ (70 mmol L$^{-1}$), $CO(NH_2)_2$ (300 mmol L$^{-1}$), and $NH_4F$ (150 mmol L$^{-1}$) in deionized water. For $Co_3O_4$-NS deposition, the precursor solution was prepared by adding $Co(NO_3)_2$ (70 mmol L$^{-1}$), $CO(NH_2)_2$ (150 mmol L$^{-1}$), and $NH_4F$ (300 mmol L$^{-1}$) in deionized water. The porous graphite felt substrate with diameter of 25 mm and thickness of 5 mm was immersed in a synthetic solution (65 mL) in a Teflon-lined stainless-steel autoclave (100 mL) and thermally treated at 120 °C for 10 h. After naturally cooling to room temperature, the $Co_3O_4$-NW or $Co_3O_4$-NS electrode was washed with deionized water and annealed at 450 °C for 2 h.

### Preparation of GO suspension
The GO suspension was prepared by adding the GO powder (40 mg) in deionized water (40 mL) with concentration of 1.0 mg mL$^{-1}$. For better dispersion, the suspension was subjected to ultrasonic treatment for exfoliation and centrifugation at 1700 g for 5 min to remove possibly undispersed powder. Then the GO suspension was diluted to 20 μg mL$^{-1}$ for nanowire perforation.

### Nanowire perforation of GO nanosheets
An electrochemical apparatus was assembled with conical inlet, Ti net cathode chamber, $Co_3O_4$-deposited porous graphite felt anode chamber, and conical outlet (Supplementary Fig. 2). Two electrodes were settled with distance of 5.0 mm, and connected with an adjustable direct-current power supply. Ti net with pore sizes around 2.0 mm was selected to alleviate the GO reduction on cathode, and porous graphite felt with pore sizes between the fibres of 50–200 μm was employed to alleviate the GO interception by $Co_3O_4$-NW anode. For nanowire perforation, the GO suspension (20 μg mL$^{-1}$) with sodium chloride (50 mmol L$^{-1}$) as electrolyte was continuously pumped into

the electrochemical apparatus with flow rate of 15 mL min[-1] and under applied voltages of 3–9 V. The GO suspension flowed through the cathode and anode sequentially. A syringe needle was inserted between the electrodes to withdraw the cathode-treated influent to analyse the solution pH.

## Preparation of GO and PGO membranes

For membrane fabrication, the GO or PGO suspension effluent (10.0 mL) from perforation apparatus was filtrated onto a poly-ethersulfone membrane with pore size of 0.22 μm by typically vacuum filtration. To obtain the membrane with different thickness, the concentration of suspension was further diluted to 10 and 5.0 μg mL$^{-1}$. Ultimately, the membrane with loading of 200 μg (14.4 μg cm$^{-2}$), 100 μg (7.2 μg cm$^{-2}$), or 50 μg (3.6 μg cm$^{-2}$) was dried at 50 °C for characterization and use.

## Separation performance evaluation

Separation performance of the prepared membranes was evaluated by using a cross-flow filtration system with effective membrane area of 7.0 cm$^2$ at room temperature. Unless otherwise specified, the feed pressure and cross-flow rate were 2.0 bar and 30.0 L h$^{-1}$ respectively. The filtration was firstly operated for 6 h to achieve stability and then collected the permeation solution and measured its volume and concentration for calculating permeance and rejection. For desalination, the Na$_2$SO$_4$ and NaCl solutions with concentration of 0.5 g L$^{-1}$ was used as feed solution. For water remediation or wastewater purification, the antibiotic or dye solutions with concentration of 100 or 50 mg L$^{-1}$ was added in to feed tank. For calculating the averaged permeance and rejection with standard deviation (error bar), the permeation data of three membrane samples were measured under the same conditions. A conductivity meter and an UV-Vis spectrophotometer (UV-1780, Shimadzu, Japan) were used to measure the concentrations of salts, antibiotics, and dyes. Rejection was calculated by the concentrations of feed and permeates. Water permeance (L m$^{-2}$ h$^{-1}$ bar$^{-1}$) was calculated based on permeate volume, effective membrane area, permeation time, and applied feed pressure. Water flux (L m$^{-2}$ h$^{-1}$) was calculated based on permeate volume, effective membrane area and permeation time. Water/Na$_2$SO$_4$ selectivity was calculated through dividing water/salt molar ratio of permeate solution by water/salt molar ratio of feed solution.

## Finite element simulation

Finite element simulation was utilized to explore the distributions of electron density, electric field strength, adsorbed OH$^-$ density over the nanowire and nanosphere on Co$_3$O$_4$ electrodes under the solution pH (11.0 and 1 mM OH$^-$ ions) of cathode-treated influent at 6.0 V. Morphological parameters of Co$_3$O$_4$-NW were set as nanowire length of 3.3 μm and radius of 20 nm, using Co$_3$O$_4$-NS with hemispherical radius of 225 nm for comparison. The relative permittivity of Co$_3$O$_4$ was set as ~3.5, and distance between the Ti net and Co$_3$O$_4$ electrodes was ~5 mm. The conductivity and relative permittivity of electrolyte (50 mM NaCl) were set as 7.5 S/cm and 70, respectively. Electric field ($E = -\nabla V$) over nanowire tips was simulated based on the opposite gradient of electric potential ($V$). Charge density ($\rho = \varepsilon_r \varepsilon_0 \nabla E$) was simulated according to Gauss's law, where $\varepsilon_O$ and $\varepsilon_r$ represent the relative permittivity parameters of electrolyte and electrode material.

## Data availability

All data in this study are included in this article and its supplementary information file. Source data are provided with this paper.

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

## Acknowledgements

This work is supported by the National Natural Science Foundation of China (Grant No. 22178143 to W.L. and Grant No. 22206058 to H.L.), Science and Technology Program of Guangzhou (Grant No. 202201020025 to W.L.), and Guangdong Basic and Applied Basic Research Foundation (Grant No. 2020B1515120036 and 2022A1515010511 to W.L.). We thank Xiaoxiong Wang at Department of Chemical and Environmental Engineering of Yale University for helping simulation.

## Author contributions

W.L. and H.L. conceived the research idea, formulated the project, and designed the detail experiments. H.L. and Y.W. carried out the electrode synthesis, nanosheet perforation, and finite element simulation. W.L., X.H., and B.K. performed nanosheet analysis, membrane assembly, separation performance evaluation, and characterizations including AFM, TEM, SEM, XPS, XRD, Raman, zeta potential, and dynamic contact angle. W.L. and H.L. processed and analysed experimental data. W.L. and H.L. wrote the manuscript. All authors revised the manuscript.

## Competing interests

The authors declare no competing interests.
