## [Peer Review File · Nature Communications]

Nanowire-assisted electrochemical perforation of graphene oxide nanosheets for molecular separationReviewers' Comments:

Reviewer #1 (Remarks to the Author):

The manuscript proposes new method to prepare perforated GO using electrochemical treatment of dispersions. The dispersions are passed through some kind of discharge cell with nanowire electrodes to make holes in GO sheets. The holey GO was then used in membranes to demonstrate effects of faster permeation due to shorter permeation pathway. Other membrane experiments include removal of dyes and antibiotics, some desalination tests. I think the study is sufficiently novel for publication in Nature Communications and in experimental part providing convincing evidence for main results. However, my impression is that quality of language, the data presentation and discussion can be significantly improved. I suggest to that authors make some major revisions to improve quality of manuscript.

1) Abstract need to be rewritten by 90%, it starts with too general and not really comprehensible sentences and miss to describe most important points of study. First sentence about “platforms” better to remove. What is “in plane accessibility of nanosheets” is also not possible to understand. I guess authors intend to say something about accessibility of space between nanosheets and transport across nanosheets. Third sentence about “interior nanopores” is also not clear as it is not defined what is that. Next sentence is not clear in the part of what is “accessible characteristics”. Much better formulations which reflect on main purpose of study need to be found. It should be very clearly formulated like “we want to make nanosheets with a lot of holes so that permeation pathway becomes much shorter”, something in that style. The same about membrane results, please formulate exact results demonstrated in the study instead of very general statements about “improving”.

2) Fig.1 reflects the idea of study quite well but I miss some semi-quantitative estimation how the holes in GO sheets improve permeation. This happen in two ways- permeation pathway is shortened (by how much approximately?) and by increasing “cross area” (area of holes is added to the space between GO sheets. Authors present AFM images showing holey structure of GO sheets. It should be possible to make some estimation for % of area in holes relative to the total area of flakes. Then one could estimate the length of pathway shortened let’s say by order of magnitude, area available for permeation by two orders or something like that. One need to cite also some information about flake sizes of original GO and processed holey GO. I note also that Fig.1 is idealized to very high degree since all GO flakes are of irregular shape and will never form accurate packing edge to edge. There will be a lot of overlaps between flakes. See for example picture from this paper: <https://doi.org/10.1021/acsnano.8b02015> The true packing of GO membranes is something closer to “voids” image but with “pinholes” added.

3) Image analysis (e.g. using ImageJ software) can possibly be used to find average hole size and even to plot hole size distribution based on AFM images. It will make model estimations suggested in 2) more subjective. Most likely smallest pores are not possible to image by AFM but in this case some rough estimations can be done using XPS and % of edge carbon atoms (% of carbons with double bonded oxygen groups) as suggested in this study: <https://doi.org/10.1021/acsami.0c11122> Relevant data are shown in Fig. 3 (o). It does shows some increase in % of edge carbons for samples treated at 3V, when larger holes are formed this % expectedly goes down somewhat. I note that very large holes are likely not to help permeation, the holes will be “filled” by neighbouring sheets. GO is rather flexible and sheets

will curve to occupy a bit larger empty space.

4) One of the most important parameters of GO materials is C/O ratio which needs to be found by using C1s and O1s. I don't find information about overall degree of oxidation. It must be added to Table 1 in SI file and discussed in the main part. The treatment performed with dispersions suppose to induce oxidation of GO sheets according to the text but exact degree of oxidation (by XPS) is not mentioned.

5) I like XRD data, which are shown for samples tested in dry and wetted state. However, I would like to see broader angle region, extend the graph to show also region up to ~60-70 degrees (at least in SI file)

6) For permeation and separation parts. Rejection in the studied membranes can be related to 2 parameters: first is size of interlayers and second is size of holes. Only small hole size help to rejection which seem to be reflected in the data. This point is not very clear from the discussion provided in the paper.

Once again, see comments 2 and 3, relate size of the holes to sizes of molecules. Also discussion about "pores" is confusing in some parts. I suggest to make clear distinction between "pore" formed by parallel sheets and "pore" as holes in GO sheets. Possibly it is better to use simply "holes" every time instead of "pore."

7) I also agree that none of the pore sizes mentioned in previous comment is sufficient to provide desalination of Na₂SO₄, also as mentioned by authors in the text.

8) "In view of the graphene thickness of 0.34 nm, the free interlayer channels of the wetted GO and PGO membranes were 0.94–1.04 nm and wider than the hydrated diameters of Na⁺ (0.71 nm) and SO₄²⁻ (0.76 nm)." Graphene thickness is not relevant to the size of interlayer channels. The thickness of GO sheets easily measured by AFM is about 0.7 nm. The "graphene capillaries" postulated in early studies of GO membranes were never confirmed experimentally and not needed to explain all experimental observations. I agree that partial dehydration is likely when ions penetrate into inter-layer space of GO membranes.

8) Some dyes are known to be sorbed by GO and especially strongly by defect-rich GO. Therefore, relevant discussion with proper citations needs to be added. For example, sorption of MB was demonstrated to correlate with amount of holes and defects in GO. The manuscript describes somewhat similar defect rich GO but obtained by different method. Dyes are likely to interact with carboxylic groups on GO which are located at defects and hole edges.

9) Language of manuscript can be improved significantly. I see many formulations which are not really precise, very long and confusing etc.

I think the paper can be improved to the level required for publication in Nature Communications after appropriate revisions. Experimental part is sufficiently well executed and interesting results are reported.

Reviewer #2 (Remarks to the Author):

In recent decades, graphene oxide (GO) nanosheets have been demonstrated as excellent building blocks to construct high-performance membranes for both water treatment and gas separation. Various

methods, such as chemical modification and perforation, and engineering of interlayer channels and GO nanosheet stacking, were explored to improve the separation performance of GO membranes. In this manuscript, a nanowire electrochemical method was proposed to perforate and modify graphene oxide nanosheets, and the as-prepared porous GO (PGO) nanosheets were assembled into membranes by frequently-used vacuum filtration. PGO membranes exhibited quadruple water permeance, and higher rejections for salts and small molecules than the original GO membranes. However, a couple of methods (<https://doi.org/10.1016/j.memsci.2021.120216>; ACS Appl. Mater. Interfaces 2020, 12, 1387–1394; RSC Adv. 2014, 4, 21425–21428; Environ. Sci. Technol. 2019, 53, 8314–8323) were already developed to drill nanopores on GO nanosheets and prepare the porous or holey GO membranes for water treatment, and a similar improvement in separation performance was observed. In addition, compared with these chemical perforation methods, the nanowire electrochemical method proposed in this manuscript does not show any advantages in efficiency and controllability. Moreover, based on the evidence presented in the manuscript, I am not convinced of the existence of a “direct reaction from losing electrons”, and I highly suspect that the perforation mechanism is mainly the indirect oxidation of GO by •OH radicals, similar to previously reported chemical methods. Therefore, I think the novelty and quality of this manuscript do not meet the high standard of Nature Communications. Here are some of my doubts and comments:

1. “nanowire perforation of GO nanosheets was conducted through two pathways of direct and indirect •OH oxidations..... For Co₃O₄-NW anode, the lower polarization potential (~0.91 V vs. Ag/AgCl) for the GO suspension than that for the solution without GO (~1.01 V vs. Ag/AgCl) validated the existence of direct GO electrochemical reaction (Fig. 2f).” The existence of GO will change the pH of the solution, leading to the polarization potential difference of OH⁻ electrochemical reaction. I think these LSV tests can't prove the existence of direct GO electrochemical reaction.
2. The separation performance of PGO membranes depends on the pore size and density. In Figure 3a-l, what are the pore size distributions and pore densities of PGO membranes?
3. In Figure 4e, it's hard to identify the XRD peaks. They should be retested to get reliable interlayer spaces.
4. GO membranes were reported to be unstable during cross-flow nanofiltration tests. Why the GO membranes in this study can stand the cross-flow nanofiltration tests and get stable water permeances and salt rejections?
5. In Figure 5d, the “permeance” should be “flux”.
6. According to the AFM and TEM images, the pore size of PGO-9 is larger than that of PGO-6. But why PGO-9 membrane has both lower permeance and salt rejection compared with PGO-6 membrane (Figure 5a,b,c)?
7. In Figure 5h, the separation performance of other previously reported porous or holey GO membranes should also be included and discussed.

8. The applied voltage determined the pore size and chemical structure of PGO. But for a chemical reaction, the reaction time is also an important parameter. What is the electrochemical oxidation time or retention time of GO in the anode chamber? And how does the retention time affect the perforation of GO?

9. Page 17, Line 1. The authors wrote: “the nanofiltration was mainly affected by Donnan effect.” And as shown in Figure 4g, the Zeta potential of the GO and PGO membranes depends on pH. So the pH and salt concentrations will affect salt and dye rejections of PGO membranes. It’s important to know the effect of pH and salt concentrations on separation performance in practical applications.

Reviewer #3 (Remarks to the Author):

Summary

o The approach mentioned in the following works is way simpler and straightforward than the presented approach:

- Buelke, C., Alshami, A., Casler, J., Lin, Y., Hickner, M. and Aljundi, I.H., 2019. Evaluating graphene oxide and holey graphene oxide membrane performance for water purification. *Journal of Membrane Science*, 588, p.117195.

- Lin, Y., Han, X., Campbell, C.J., Kim, J.W., Zhao, B., Luo, W., Dai, J., Hu, L. and Connell, J.W., 2015. Holey graphene nanomanufacturing: Structure, composition, and electrochemical properties. *Advanced Functional Materials*, 25(19), pp.2920-2927.

is probably a better method for hole production: more controlled, smaller holes.

o Concerns that all GO was being converted into PGO via the nanowires. What is more likely is that only some of the GO was actually reacted, and that some of those reactions produced PGCl since the electrolyte solution this was being done in consisted of 2.9 mg/mL NaCl.

o Linear scanning voltammetry is not a good method for measuring PGO production in this setup

o Inaccurate claims about XPS data

o Low Na₂SO₄ concentrations used, and no data shown for NaCl tests besides < 40% rejection

o No control (bare PES) membrane test! Strongly suggested that they do this

o Full read for grammar

Abstract

o “It remains great challenging to design nanosheets with uniform interior nanopores.”

♣ Lin 2015 showed that open air tube furnace heating at high temps over a short or long period of time produced holey graphene with small or large, but uniform, holes.

Method

o Did not list ultrasonication time for exfoliating GO in water, although we can assume that it was well

suspended. Also centrifugation/decantation to remove unexfoliated material, but didn't list duration at 4000 RPM.

o For the nanowire perforation of the GO sheets, I have serious concerns about the yield of perforated GO (PGO). The nanowire electrode apparatus has a diameter of 5 mm, with flow velocity of 12.5 mm/s. So relative to the reaction chamber, the fluid is moving at a decent pace. However, the nanowires themselves are only 3 microns long along the edge of the chamber, so 99.94% of the reaction chamber is flowing bulk fluid. While the anode will attract GO to the nanowires via electrophoresis, I wonder how much actual PGO is formed. Assuming the perforation is instantaneous, the question becomes "how much GO migrates to the nanowires and then leaves per second?" My guess is, while the majority of GO is indeed perforated, there remains quite a bit that exits the chamber unreacted.

o Additionally, the concentration of GO used in this process is 0.02 mg/mL. However, it was not done in DI water, and instead they used NaCl as the electrolyte solution during the perforation process. The NaCl concentration was 2.9 mg/mL, so I also have to wonder now that, not only "how much GO exited the chamber unreacted", but "how much GO was instead converted to graphene chloride (GCl)?" They make the case that electrolyzed water from the cathode can form OH⁻, which can then attach to the nanowire tips at the anode to reduce to OH[·], so they are aware that other side reactions are occurring and material other than GO are migrating to the nanowires.

o Along those same lines, they measure the reaction occurrence at the nanowire tip via linear scanning voltammetry, and they equate this to PGO production: "there's a change in voltage, so PGO is being produced." However, since there is much more activity at the nanowire tip other than the PGO reaction that would change the voltage, it's not a reliable metric for PGO production.

Characterization

o For AFM graphs, they state that GO was monolayer at ~1 nm, and that PGO was thicker at around 1.8 nm. They suggest that this increase in thickness is due to "grafted out-of-plane functional groups", but the XPS data shows little change in functional groups. It most likely instead is a bilayer of PGO. Perhaps the perforation reaction can cause some GO sheets to crosslink

o Speaking of XPS data, they state that the C 1s peak areas were expanded, and that the COOH and C=O peaks for the O 1s graphs were more intense. Neither show much change for PGO6 and PGO9, with some change for PGO3. from what I can tell. Figures 3n and 3o also confirm this?

o For XRD, they don't list the PGO9 wet state d-spacing. However, from the graph, we can tell it would be more than 1.38 nm. Large increases seen here, even for their GO at 0.91 nm to 1.28 nm (we had an increase of 0.71 nm to 0.85 nm). Since they used such low loading on the membrane during vacuum filtration, my guess is that what little GO or PGO was there had an easier time to expand since there wasn't additional GO to keep it from expanding: sort of like "GO compressing GO", or self-compression.

Performance

o Testing parameters are 500 mL/min at 2 bar for the cross-flow cell, 6 hours of compression before data collection, and concentrations of 50 mg/L for methyl blue, 0.5 mg/mL for Na₂SO₄, and unknown concentration for NaCl.

o They never did a control test for seeing PES membrane performance. This is critical! The large d-spacings when wet, combined with the low loading, leads me to believe the support membrane is doing a lot of the heavy lifting, and a control test would prove otherwise.

o Supp Figure 17 shows PGO3 had best separation factor of water/Na₂SO₄ across all loadings. PGO6 had

highest permeance. This makes sense since PGO3 had no holes and slightly increased oxygen-containing functional groups, and PGO6 had holes, which were smaller than PGO9. Small holes are better than big holes? Yet another reason that open air tube furnace processing is better in my opinion: the entire sample undergoes controlled oxidation to form small holes.

o Low rejection of NaCl at < 40%, and no other data presented. Since PGO3 had the best rejection of Na₂SO₄, we can assume 40% rejection of NaCl was achieved with that membrane.

**Response letter for “Nanowire electrochemical perforation of graphene oxide
nanosheets for membrane separation (NCOMMS-23-35238-T)”**

Reviewers' comments:

Reviewer #1 (Remarks to the Author):

The manuscript proposes new method to prepare perforated GO using electrochemical treatment of dispersions. The dispersions are passed through some kind of discharge cell with nanowire electrodes to make holes in GO sheets. The holey GO was then used in membranes to demonstrate effects of faster permeation due to shorter permeation pathway. Other membrane experiments include removal of dyes and antibiotics, some desalination tests. I think the study is sufficiently novel for publication in Nature Communications and in experimental part providing convincing evidence for main results. However, my impression is that quality of language, the data presentation and discussion can be significantly improved. I suggest to that authors make some major revisions to improve quality of manuscript.

Response/Action:

We thank the reviewer for his/her positive comments.

1) Abstract need to be rewritten by 90%, it starts with too general and not really comprehensible sentences and miss to describe most important points of study. First sentence about “platforms” better to remove. What is “in plane accessibility of nanosheets” is also not possible to understand. I guess authors intend to say something about accessibility of space between nanosheets and transport across nanosheets. Third sentence about “interior nanopores” is also not clear as it is not defined what is that. Next sentence is not

clear in the part of what is “accessible characteristics”. Much better formulations which reflect on main purpose of study need to be found. It should be very clearly formulated like “we want to make nanosheets with a lot of holes so that permeation pathway becomes much shorter”, something in that style. The same about membrane results, please formulate exact results demonstrated in the study instead of very general statements about “improving”.

Response/Action:

We thank the reviewer for his/her comments. We have removed the first sentence about “platforms” and revised the sentences about “in plane accessibility of nanosheets, interior nanopores, and accessible characteristics” in the manuscript. We have formulated the exact results of the membrane separation performance and rewritten the abstract as follow.

“Two-dimensional nanosheets, e.g., graphene oxide (GO), have been widely used to fabricate efficient membranes for molecular separation. However, because of poor transport across nanosheets and high width-to-thickness ratio, the permeation pathway length and tortuosity of these membranes are extremely large, which limit their separation performance. Here we report a facile, scalable, and controllable nanowire electrochemical concept for perforating and modifying nanosheets to shorten permeation pathway and adjust transport property. It is found that confinement effects with locally enhanced charge density, electric field, and hydroxyl radical generation over nanowire tips on anode can be executed under low voltage, thereby inducing confined direct electron loss and indirect oxidation to reform configuration and composition of GO nanosheets. We demonstrate that the porous GO nanosheets with a lot of holes are suitable for assembling separation membranes with tuned accessibility, tortuosity, interlayer space, electronegativity, and hydrophilicity. For molecular separation, the prepared membranes exhibit

quadruple water permeance and higher rejections for salts (> 91%) and small molecules (> 96%) as/than original ones. This nanowire electrochemical perforation concept offers a feasible strategy to reconstruct two-dimensional materials and tune their transport property for separation.”

2) Fig.1 reflects the idea of study quite well but I miss some semi-quantitative estimation how the holes in GO sheets improve permeation. This happen in two ways- permeation pathway is shortened (by how much approximately?) and by increasing “cross area” (area of holes is added to the space between GO sheets. Authors present AFM images showing holey structure of GO sheets. It should be possible to make some estimation for % of area in holes relative to the total area of flakes. Then one could estimate- the length of pathway shortened let’s say by order of magnitude, area available for permeation by two orders or something like that. One need to cite also some information about flake sizes of original GO and processed holey GO. I note also that Fig.1 is idealized to very high degree since all GO flakes are of irregular shape and will never form accurate packing edge to edge. There will be a lot of everlaps between flakes. See for example picture from this paper: <https://doi.org/10.1021/acsnano.8b02015> The true packing of GO membranes is something closer to “voids” image but with “pinholes” added.

Response/Action:

We thank the reviewer for his/her comments. We have made the estimation for the area ratio of holes in nanosheets by using ImageJ software based on the AFM image as mentioned in comment 3. The result indicated that the area ratio of holes of PGO-6 was 37.4%. Because of the overlapping, it was difficult to measure the area ratio of PGO-9. We have also counted the size distribution of holes in PGO. As shown in Supplementary Fig. 6, the PGO-6 had uniform holes with average diameter of 35 nm. Based on this, the

density of holes in PGO-6 could be approximately estimated as $4.0 \times 10^{14} \text{ m}^{-2}$.

For permeation pathway of GO membranes, the molecules have to pass through vertical defects/edges and shuttle back and forth in horizontal interlayer channels between adjacent nanosheets. The transport pathway is long and the tortuosity is extremely large. The transport pathway length (S) of GO membranes could be calculated based on membrane thickness (h), interlayer space (d), and nanosheet size (L , assumed as $1 \mu\text{m}$) through an equation of $S=h/d \times L$ (*Environ. Sci. Technol.* **53**, 8314-8323 (2019); *ACS Appl. Mater. Interfaces* **12**, 1387-1394 (2020)). For PGO membranes, because the holes could serve as vertical channels, the length of transport pathway could be estimated as $S=h/d \times L / ((L^2 \times \rho)^{0.5} + 1)$. ρ was the density of holes. Based on these two equations, it was calculated that the transport pathway length of the PGO-6 membrane was only about 5% as that of GO. In other words, the tortuosity of the PGO-6 membrane was one twentieth as that of GO. Compared with the reductions in pathway length and tortuosity, the quadruple permeance increment of the PGO-6 membrane as the GO membrane was relatively lower. This might be resulted from that the pinhole defects in the GO and PGO nanosheets and the voids from a certain irregular stacking could also serve as vertical channels and shorten the pathway length to increase the water permeance of the GO and PGO membranes (*ACS Nano* **12**, 7855-7865 (2018)), unlike an ideal GO membrane stacked at high degree and with no defect.

For Fig. 1, we agree with the comments of the reviewer. Fig. 1 was used to describe the main point of this manuscript that nanowire electrochemical perforation of GO nanosheets could improve separation performance of membranes. Therefore, we have added Supplementary Fig. 1 to describe the more true packing of GO membranes with voids and pinholes and the transport property of the GO and PGO membranes (*ACS Nano* **12**, 7855-7865 (2018)).

We have revised the manuscript. Some discussions and figures about hole property, permeation

pathway, separation mechanism, and GO membrane stacking have been added in the revised manuscript as follow. The related references have also been cited and discussed.

“It should be noted that many uniform holes were generated in the PGO-6 and PGO-9 nanosheets and became larger as voltage raised, with average diameters of 35 and 85 nm, respectively (Supplementary Fig. 6). Considering the 37.4% area ratio of holes from ImageJ, the density of holes in PGO-6 could be approximately estimated as $4.0 \times 10^{14} \text{ m}^{-2}$.”

“For GO and PGO membranes, the transport pathway length (S) could be calculated based on membrane thickness (h), interlayer space (d), nanosheet size (L , assumed as $1 \mu\text{m}$), and hole density (ρ) as equations of $S=h/d \times L$ and $S=h/d \times L/((L^2 \times \rho)^{0.5} + 1)$, respectively^{49,51}. It was calculated that the transport pathway length of PGO-6 was only about 5% as that of GO. In other words, the tortuosity of the PGO-6 membrane was one twentieth as that of GO. Relative to the reduction in pathway length, the quadruple permeance increment of PGO-6 as GO was lower. This might be resulted from that the pinhole defects in the GO and PGO nanosheets and the free spaces from a certain irregular stacking could serve as channels to increase the water permeance of the GO and PGO membranes⁵⁸.”

Supplementary Fig. 1. Schematics of transport pathway of the conventional GO membranes and porous

GO membranes with holes⁵⁸. Black lines, grey dashed boxes, white dashed boxes, and dark cyan arrows represent GO nanosheets, pinhole defects, perforated holes, and water transport pathways of the GO and PGO membranes. Some wrinkles and voids from a certain irregular stacking may affect molecular transport of GO membranes.

Supplementary Fig. 6. Size distributions of holes in the PGO nanosheets based on the AFM images.

49. Chen, X. et al. Reduced holey graphene oxide membranes for desalination with improved water permeance. *ACS Appl. Mater. Interfaces* **12**, 1387-1394 (2020).

51. Li, Y. et al. Thermally reduced nanoporous graphene oxide membrane for desalination. *Environ. Sci. Technol.* **53**, 8314-8323 (2019).

58. Saraswat, V. et al. Invariance of water permeance through size-differentiated graphene oxide laminates. *ACS Nano* **12**, 7855-7865 (2018).

3) Image analysis (e.g. using ImageJ software) can possibly be used to find average hole size and even to plot hole size distribution based on AFM images. It will make model estimations suggested in 2) more subjective. Most likely smallest pores are not possible to image by AFM but in this case some rough estimations can be done using XPS and % of edge carbon atoms (% of carbons with double bonded oxygen groups) as suggested in this study: <https://doi.org/10.1021/acsami.0c11122> Relevant data are shown in Fig. 3 (o). It does shows some increase in % of edge carbons for samples treated at 3V, when larger holes are formed this % expectedly goes down somewhat. I note that very large holes are likely not to help permeation, the holes will be “filled” by neighbouring sheets. GO is rather flexible and sheets will curve to occupy a bit larger empty space.

Response/Action:

We thank the reviewer for his/her comments. As mentioned in the response of comment 2. We have estimated the area ratio of holes (37.4%) in the PGO-6 nanosheets by using ImageJ software based on the AFM image. We have also measured the average size of holes and its size distribution by using Nano Measurer software. The PGO-6 nanosheet had uniform holes with average diameter of 35 nm. Base on the area ratio and average hole size, the hole density of PGO-6 could be estimated as $4.0 \times 10^{14} \text{ m}^{-2}$.

As reviewer's comments, it is not possible to image the “smallest pores” (pinhole defects) in nanosheets by AFM, but the defects can be roughly estimated by using the content of edge carbon atoms with double bonded oxygen groups from the XPS data (*ACS Appl. Mater. Interfaces* **12**, 45122-45135 (2020)). As the results, the area ratio of “smallest pores” in GO, PGO-3, PGO-6, and PGO-9 could be estimated as 8.3%, 16.0%, 11.8%, and 12.8%, respectively.

We agree with the comment that the very large holes are likely not to help permeation, because the

holes will be “filled” by neighbouring sheets and the flexible GO sheets will curve to occupy large empty space. The experimental results of the membrane separation were consistent with this comment. The PGO-9 membrane with larger holes showed poorer rejection and permeance than PGO-6 due to the excessive irregularity and nanosheet damage degree.

We have revised the manuscript. The discussions about defects and membrane permeation have been added in the revised manuscript.

“As reported in previous study⁴⁴, the pinhole defects that were not AFM-detectable could be roughly estimated by using the edge carbon atoms with double bonded oxygen. So the area ratio of pinhole defects in GO, PGO-3, PGO-6, and PGO-9 were approximately 8.3%, 16.0%, 11.8%, and 12.8%, respectively.”

“Less permeance upgradation of PGO-3 than PGO-6 was explained by their inferior porosity; while the poorer permeance of PGO-9 than PGO-6 was attributed to the excessive membrane irregularity from nanosheet damage and the occupation of neighbouring nanosheets to large space of holes.”

44. Boulanger, N. et al. Enhanced sorption of radionuclides by defect-rich graphene oxide. *ACS Appl. Mater. Interfaces* **12**, 45122–45135 (2020).

4) One of the most important parameters of GO materials is C/O ratio which needs to be found by using C1s and O1s. I don't find information about overall degree of oxidation. It must be added to Table 1 in SI file and discussed in the main part. The treatment performed with dispersions suppose to induce oxidation of GO sheets according to the text but exact degree of oxidation (by XPS) is not mentioned.

Response/Action:

We thank the reviewer for his/her comments. We have added the overall oxidation degree (C/O ratio) of GO and PGO in Supplementary Table 1. The GO, PGO-3, PGO-6, and PGO-9 membranes had C/O ratio of 2.46, 1.96, 2.34, and 2.25, respectively.

Supplementary Table 1. C/O ratio, C-H/C-C/C=C, C-OH/C-O-C, C=O, and COOH contents of the GO and PGO nanosheets from C 1s and O 1s XPS spectra.

Nanosheet	C/O ratio	C-H/C-C/C=C	C-OH/C-O-C	C=O	COOH
GO	2.46	60.26	31.44	5.85	2.45
PGO-3	1.96	48.84	35.20	10.82	5.14
PGO-6	2.34	57.63	30.61	8.47	3.29
PGO-9	2.25	57.05	30.10	9.85	3.00

5) I like XRD data, which are shown for samples tested in dry and wetted state. However, I would like to see broader angle region, extend the graph to show also region up to ~60-70 degrees (at least in SI file)

Response/Action:

We thank the reviewer for his/her comments. We will recollect the XRD data in the revised manuscript. In most studies, the XRD data of GO membranes were collected from 5° to up to 40° (usually 20°–40°). We have collected the XRD data in the range of 5°–40°. As shown in Supplementary Fig. 15, the broader peaks between 15°–30° were assigned to the polymer substrates.

Supplementary Fig. 15. XRD patterns of the GO, PGO-3, PGO-6, and PGO-9 membranes at dry and wetted states.

6) For permeation and separation parts. Rejection in the studied membranes can be related to 2 parameters: first is size of interlayers and second is size of holes. Only small hole size help to rejection which seem to be reflected in the data. This point is not very clear from the discussion provided in the paper.

Response/Action:

We thank the reviewer for his/her comments. As reviewer’s comments, the sizes of holes and interlayers affect rejection. For holes, the holes with size precisely located between the diameters of water and salts can play a role for rejection. This precise size may be assigned to the some special defects of GO nanosheets. Such kind of defects may exist but should be in all GO, PGO-3, PGO-6, and PGO-9 membranes, identified from the XPS results. Therefore, we have not attributed to the rejection improvement of the PGO membranes by such kind of pinhole defects, though the most pinhole defects in the PGO-3 membranes might play a role for the highest rejection among four membranes.

For interlayer space, as mentioned in the manuscript, because the size of interlayer space of the membranes was usually larger than the hydrated diameters of Na^+ and SO_4^{2-} , thus we did not think that the interlayer space could sieve the salt ions from water. Therefore, we believed that the size exclusion might be not a dominated mechanism for Na_2SO_4 rejection, especially considering the peeling of water from the shell of hydrated ions.

Strong electrostatic repulsion of negatively charged surface towards SO_4^{2-} endowed the membranes with good rejection. Consequently, the maximum rejection was achieved by the PGO-3 membrane with largest electronegativity. It should be pointed out that the slightly smaller zeta potential of PGO-6 than GO at pH of ~7 seemed difficult to bring about substantial rejection amelioration. We calculated the permeation rates and water/salt selectivity of various membranes with different thickness under various feed pressures (Fig. 5f and Supplementary Fig. 23). Corresponding to permeance and rejection, the water permeation rate and water/salt selectivity had analogous changing trend with the membrane thickness and perforation voltage. For instance, the selectivity of the PGO-3 membrane was an order of magnitude as that of the GO membrane, up to 29.5. Differently, the Na_2SO_4 permeation rate of various membranes was roughly ordered by $\text{PGO-3} < \text{GO} < \text{PGO-6} < \text{PGO-9}$. A similar or lower Na_2SO_4 permeation rate of PGO-6 than PGO-9 was attributed to that the more regular stacking provided more uniform electronegative sub-nanometre interlayer channels for better electrostatic repulsion. Because the increment of water permeation rate was greater than that of Na_2SO_4 , the PGO-6 membrane had better rejection and selectivity. In other words, the water permeation enhancement significantly contributed to the rejection and selectivity of the PGO-6 membrane. We have revised the related parts of the manuscript.

Once again, see comments 2 and 3, relate size of the holes to sizes of molecules. Also discussion about

“pores” is confusing in some parts. I suggest to make clear distinction between “pore” formed by parallel sheets and “pore” as holes in GO sheets. Possibly it is better to use simply “holes” every time instead of “pore.”

Response/Action:

We thank the reviewer for his/her comments. We have discussed the size of the holes in comments 2 and 3. Relative to the size of molecules, the size of holes in PGO-6 were larger. Because both the GO and PGO-6 membranes showed high rejections for all neutral, electronegative, and electropositive antibiotic/dye molecules, and the organic molecules had relatively larger diameters than interlayer space of membranes, we believed that the size exclusion from the interlayer space was critical to antibiotic and dye removals.

We have made the clear distinction for the different pores. The pores formed by parallel nanosheets, formed by carbon-carbon fracture during oxidation of graphene and nanowire electrochemical perforation with ultrasmall size, and perforated by nanowire electrochemical method with AFM-detectable size were described as interlayer channels (spaces), pinhole defects, and holes, respectively.

7) I also agree that none of the pore sizes mentioned in previous comment is sufficient to provide desalination of Na_2SO_4 , also as mentioned by authors in the text.

Response/Action:

We thank the reviewer for his/her comments. We did think that the desalination of Na_2SO_4 was mainly affected by Donnan effect and the increment differences between water and Na_2SO_4 permeation rates from nanowire electrochemical perforation and modification, as mentioned in the revised manuscript and the

response of comment 6.

8) “In view of the graphene thickness of 0.34 nm, the free interlayer channels of the wetted GO and PGO membranes were 0.94–1.04 nm and wider than the hydrated diameters of Na⁺ (0.71 nm) and SO₄²⁻ (0.76 nm).” Graphene thickness is not relevant to the size of interlayer channels. The thickness of GO sheets easily measured by AFM is about 0.7 nm. The “graphene capillaries” postulated in early studies of GO membranes were never confirmed experimentally and not needed to explain all experimental observations. I agree that partial dehydration is likely when ions penetrate into inter-layer space of GO membranes.

Response/Action:

We thank the reviewer for his/her comments. We agree with the comments of the reviewer. Indeed, the “graphene capillaries” postulated in early studies were never confirmed experimentally. In fact, in this study, the desalination of Na₂SO₄ was affected by Donnan effect, rather than the size exclusion from interlayer channels. Therefore, we have removed the related explanation as per the reviewer’s comments.

8) Some dyes are known to be sorbed by GO and especially strongly by defect-rich GO. Therefore, relevant discussion with proper citations needs to be added. For example, sorption of MB was demonstrated to correlate with amount of holes and defects in GO. The manuscript describes somewhat similar defect rich GO but obtained by different method. Dyes are likely to interact with carboxylic groups on GO which are located at defects and hole edges.

Response/Action:

We thank the reviewer for his/her comments. We agree with that the dyes are likely to interact with carboxylic groups on GO. In this study, a cross-flow nanofiltration apparatus was used to evaluate the separation performance. So the adsorption of dye on GO could be almost ignored for rejection. For the filtration system with three membrane samples and 1000 mL of dye solution, the filtration was circulated for 6 h to achieve stability and then the permeate solution was collected to measure permeance and rejection. During this period of 6-h stabilization, over 1500 mL of permeate solution passed through three membrane samples. If the adsorption greatly contributed to the rejection, the dye solution concentration in the feed tank should be reduced greatly and almost all dye molecules should be adsorbed by the membranes with adsorption capacity up to 333 mg mg^{-1} (50 mg dye in 1000 mL for three membranes with PGO loading of 50 μg). However, the dye solution concentration became slightly higher at $55\text{--}60 \text{ mg L}^{-1}$ (neutral red, varied at different experiments), due to the possible loss and sample collection of permeate solution during experiments. Therefore, the adsorption should not be a main factor for separation.

We have revised the manuscript and the added the related discussion.

“It should be noted that the dye solution concentration became slightly higher after filtration ($55\text{--}60 \text{ mg L}^{-1}$ varied for different experiments) due to the loss and sample collection of permeate solution, suggesting the separation was based on rejection, rather than adsorption.”

9) Language of manuscript can be improved significantly. I see many formulations which are not really precise, very long and confusing etc.

Response/Action:

We thank the reviewer for his/her comments. We have improved the language of manuscript.

I think the paper can be improved to the level required for publication in Nature Communications after appropriate revisions. Experimental part is sufficiently well executed and interesting results are reported.

Reviewer #2 (Remarks to the Author):

In recent decades, graphene oxide (GO) nanosheets have been demonstrated as excellent building blocks to construct high-performance membranes for both water treatment and gas separation. Various methods, such as chemical modification and perforation, and engineering of interlayer channels and GO nanosheet stacking, were explored to improve the separation performance of GO membranes. In this manuscript, a nanowire electrochemical method was proposed to perforate and modify graphene oxide nanosheets, and the as-prepared porous GO (PGO) nanosheets were assembled into membranes by frequently-used vacuum filtration. PGO membranes exhibited quadruple water permeance, and higher rejections for salts and small molecules than the original GO membranes. However, a couple of methods (<https://doi.org/10.1016/j.memsci.2021.120216>; ACS Appl. Mater. Interfaces 2020, 12, 1387-1394; RSC Adv. 2014, 4, 21425-21428; Environ. Sci. Technol. 2019, 53, 8314-8323) were already developed to drill nanopores on GO nanosheets and prepare the porous or holey GO membranes for water treatment, and a similar improvement in separation performance was observed. In addition, compared with these chemical perforation methods, the nanowire electrochemical method proposed in this manuscript does not show any advantages in efficiency and controllability. Moreover, based on the evidence presented in the manuscript, I am not convinced of the existence of a “direct reaction from losing electrons”, and I highly suspect that the perforation mechanism is mainly the indirect oxidation of GO by •OH radicals, similar to previously reported chemical methods. Therefore, I think the novelty and quality of this manuscript do not meet the high standard of Nature Communications. Here are some of my doubts and comments:

Response/Action:

We thank the reviewer for his/her comments. It is possible that we have not comprehensively discussed the

characteristics of nanowire electrochemical method for perforating and modifying GO nanosheets, thereby making that the reviewer unable to clearly understand the advantages in efficiency and controllability and the perforation mechanism of this method.

For the advantages, nanowire electrochemical method had some advantages in operation, fabrication condition, environmental friendliness, economy, safety, efficiency, controllability, and GO property, compared with conventional methods. 1) Unlike the intermittent batch processing methods mentioned by the reviewer, e.g., chemical etching by H_2O_2 , the nanowire electrochemical method had continuous flow-through process, which might be more beneficial to the operability, controllability, and scalability of preparation and production. 2) Relative to the used reagents of NH_4OH , H_2O_2 , KMnO_4 , oxalic acid, hydrochloric acid, and N-methylpyrrolidone or the high temperature of $\sim 400\text{ }^\circ\text{C}$, the NaCl and room temperature of nanowire electrochemical method might be benefit for the environmentally friendly, mild, economical, and safe production of PGO. 3) For the perspective of efficiency, the hydraulic retention time of GO suspension in the porous anode was only 9.8 s for nanowire electrochemical method. The flux of electrochemical apparatus for GO perforation was up to $1834\text{ L m}^{-2}\text{ h}$. Compared the processing time for batch processing methods usually having duration of several hours, the processing efficiency might also be an advantage. 4) As for GO property and reaction controllability, the chemical etching by H_2O_2 possibly involved the etching reaction of oxygenated regions in GO, thus possibly inducing the reduction of GO, which might be a negative factor for membrane permeance. Comparatively, because of the strong electrochemical reactions over nanowire tips, the nanowire electrochemical method could perform oxidation and perforation for both graphitic and oxygen-containing regions, thus causing the formation of PGO nanosheets with even higher oxygen contents and uniform holes. Moreover, the size of holes could be controlled by adjusting applied voltages.

For the perforation mechanism, the reviewer was not convinced of the existence of “direct reaction from losing electrons” and think that the perforation mechanism is mainly the indirect oxidation of GO by •OH radicals, similar to previously reported chemical methods. Nanowire anode could perform confinement effects with concentrated positive charge over their tips, through driving the migration of free electrons away from their tips by low external voltage (< 10 V). The Co₃O₄-NW anode possessed the locally charge density up to 10⁷ C m⁻³ by finite element simulations, which were about three orders of magnitude as those of Co₃O₄-NS. Such high finite-concentrated reaction activity over nanowire tips could significantly facilitate the adsorption and direct oxidation of negatively charged GO. Moreover, as mentioned above, previous works suggested that hydroxyl radicals preferred to attack carbon atoms where hydroxyl and epoxy groups were bonded to, which usually resulted in the loss of oxygen-containing groups on GO. However, our results showed an increase in oxygen-containing groups after nanowire perforation. In addition, we also employed LSV to recognize the occurrence of direct and indirect oxidation reactions of GO on nanowire anode (see more details in the Response/Action of comment 1).

We have added the Supplementary Table 2 about the comparison among different methods and the related discussion for better understanding the efficiency and controllability of nanowire electrochemical method for perforating and modifying GO nanosheets.

“So far, some other methods, e.g., chemical etching and air oxidation, have also been reported for generation of holes in GO nanosheets (Supplementary Table 2)⁴⁸⁻⁵³. Compared with hydrogen peroxide-needed chemical etching and high-temperature air oxidation, which had intermittent batch processing for several hours, the nanowire electrochemical strategy reported here, with continuous flow-through procedures, room temperature condition, and GO suspension flux over 1800 L m⁻² h, might have superiorities in operability, controllability, scalability, environmental friendliness, economy, safety,

and efficiency. Moreover, relative to the previously reported methods, which might involve the reactions at oxygen-containing regions of GO or at defect regions of graphene, the nanowire electrochemical method could perform oxidation and perforation for both graphitic and oxygen-containing regions because of the strong electrochemical reactions over nanowire tips.”

Supplementary Table 2. Comparison between nanowire electrochemical perforation and other methods.

RT: Room temperature; and NMP: N-methylpyrrolidone.

Method	Operation	Reagent	Condition	Raw	C/O	Ref.
Chemical etching	Batch	NH_4OH and H_2O_2	50 °C for 1–5 h	Liquid	Higher	48
Chemical etching	Batch	H_2O_2	100 °C for 1–4 h	Liquid	-	49
Chemical etching	Batch	$KMnO_4$, oxalic acid, and hydrochloric acid	RT for ~6 h	Liquid	Similar	50
Chemical etching	Batch	H_2O_2	70 °C for 10 h	Liquid	Similar	51
Air oxidation processing graphene and oxidation	Batch	Air	395–460 °C for 10 h	Solid	-	52,53
Thermal annealing and microwave treatment	Batch	Air and NMP	200 °C for 10 min	Solid	Higher	27
Nanowire	Flow-thro	NaCl	RT	Liquid	Lower or	This

References

48. Wu, T., Moghadam, F. & Li, K. High-performance porous graphene oxide hollow fiber membranes with tailored pore sizes for water purification. *J. Membr. Sci.* **645**, 120216 (2022).
49. Chen, X. et al. Reduced holey graphene oxide membranes for desalination with improved water permeance. *ACS Appl. Mater. Interfaces* **12**, 1387-1394 (2020).
50. Ying, Y., Sun, L., Wang, Q., Fan, Z. & Peng, X. In-plane mesoporous graphene oxide nanosheet assembled membranes for molecular separation. *RSC Adv.* **4**, 21425-21428 (2014).
51. Li, Y. et al. Thermally reduced nanoporous graphene oxide membrane for desalination. *Environ. Sci. Technol.* **53**, 8314-8323 (2019).
52. Buelke, C. et al. Evaluating graphene oxide and holey graphene oxide membrane performance for water purification. *J. Membr. Sci.* **588**, 117195 (2019).
53. Lin, Y. et al. Holey graphene nanomanufacturing: structure, composition, and electrochemical properties. *Adv. Funct. Mater.* **25**, 2920-2927 (2015).
27. Kang, J. et al. Microwave-assisted design of nanoporous graphene membrane for ultrafast and switchable organic solvent nanofiltration. *Nat. Commun.* **14**, 901 (2023).

1. “nanowire perforation of GO nanosheets was conducted through two pathways of direct and indirect •OH oxidations..... For Co₃O₄-NW anode, the lower polarization potential (~0.91 V vs. Ag/AgCl) for the GO suspension than that for the solution without GO (~1.01 V vs. Ag/AgCl) validated the existence of

direct GO electrochemical reaction (Fig. 2f).” The existence of GO will change the pH of the solution, leading to the polarization potential difference of OH⁻ electrochemical reaction. I think these LSV tests can't prove the existence of direct GO electrochemical reaction.

Response/Action:

We thank the reviewer for his/her comments. As the reviewer's comments, the existence of GO will change the pH of the solution, leading to the polarization potential difference of OH⁻ electrochemical reaction. The acidic substances from GO would neutralize the OH⁻ ions in cathode-treated influent, and the lower OH⁻ concentrations theoretically required larger anode potential (>1.01 V vs. Ag/AgCl) for OH⁻ ions oxidation to •OH radicals. However, as compared with influent solution without GO, the polarization potential (~0.91 V vs. Ag/AgCl) for Co₃O₄-NW anode was lower for the GO suspension, which strongly supported the occurrence of reaction at low anode potentials. The above results suggested that direct GO electrochemical reaction on Co₃O₄-NW. We have added above statement in the manuscript.

We have revised the related part of the manuscript.

“Theoretically, the acidic substances of GO nanosheets would neutralize the OH⁻ of cathode-treated influent and then increase the electrode potential for OH⁻ oxidation. However, the lower polarization potential (~0.91 V vs. Ag/AgCl) of Co₃O₄-NW for the GO suspension than that for the solution without GO (~1.01 V vs. Ag/AgCl) was observed. This phenomenon validated the existence of direct GO electrochemical reaction (Fig. 2f).”

2. The separation performance of PGO membranes depends on the pore size and density. In Figure 3a-l, what are the pore size distributions and pore densities of PGO membranes?

Response/Action:

We thank the reviewer for his/her comments. We have made the estimation for area ratio of holes and pores in flakes by using ImageJ software and XPS data. For the GO and PGO nanosheets, there were many pinhole defects from GO fabrication and nanowire electrochemical perforation, which were not detectable by AFM. These pinhole defects could be roughly estimated by using the contents of edge carbon atoms with double bonded oxygen groups (*ACS Appl. Mater. Interfaces* **12**, 45122-45135 (2020)). The defect area ratio of GO, PGO-3, PGO-6, and PGO-9 could be roughly estimated as 8.3%, 16.0%, 11.8%, and 12.8%, respectively. Besides pinhole defects, we have made the estimation for area ratio of holes in nanosheets by using ImageJ software and measured the average size of holes and its size distribution by using Nano Measurer software based on the AFM images. The results indicated that the PGO-6 had uniform holes with average diameter of 35 nm and area ratio of holes of 37.4%. Based on this, the density of the holes in PGO-6 could be roughly estimated as $4.0 \times 10^{14} \text{ m}^{-2}$.

We have revised the related parts of the manuscript. Some discussions and figures about pore size and density have been added in the revised manuscript as follow.

“It should be noted that many uniform holes were generated in the PGO-6 and PGO-9 nanosheets and became larger as voltage raised, with average diameters of 35 and 85 nm, respectively (Supplementary Fig. 6). Considering the 37.4% area ratio of holes from ImageJ, the density of holes in PGO-6 could be approximately estimated as $4.0 \times 10^{14} \text{ m}^{-2}$.”

“As reported in previous study⁴⁴, the pinhole defects that were not AFM-detectable could be roughly estimated by using the edge carbon atoms with double bonded oxygen. So the area ratio of pinhole defects in GO, PGO-3, PGO-6, and PGO-9 were approximately 8.3%, 16.0%, 11.8%, and 12.8%, respectively.”

Supplementary Fig. 6. Size distributions of holes in the PGO nanosheets based on the AFM images.

3. In Figure 4e, it's hard to identify the XRD peaks. They should be retested to get reliable interlayer spaces.

Response/Action:

We thank the reviewer for his/her comments. For the thin GO membranes, the XRD peak intensity is usually weak (*Nat. Commun.* **14**, 1016 (2023)). Although the thick GO membranes may have intensive peak, the stacking of the GO nanosheets far away the substrates are different from those close to substrates. So the XRD characteristics of the thick membranes may not be able to accurately reflect the property of the thin GO membranes. If the reviewer think that the XRD data with higher peak intensity are necessary, we will added the XRD data of the thick GO membranes.

4. GO membranes were reported to be unstable during cross-flow nanofiltration tests. Why the GO membranes in this study can stand the cross-flow nanofiltration tests and get stable water permeances and salt rejections?

Response/Action:

We thank the reviewer for his/her comments. We agree with the comments that some GO membranes were reported to be unstable during cross-flow nanofiltration. In fact, the thick GO membranes might be unstable in filtration and would detach from the substrates and then disperse in solutions. However, the ultrathin GO membranes could be employed for cross-flow nanofiltration, even under more harsh condition for cross-flow reverse osmosis (*Nat. Commun.* **14**, 1016 (2023)). Moreover, the GO and PGO suspensions for membrane fabrication contained sodium chloride and the prepared membranes had sodium and chlorine contents of 3.0–3.5% and 1.5–1.9%, respectively (Supplementary Fig. 9,10 in the revised manuscript). As demonstrated in previous studies (*Nature* **550**, 380-383 (2017)), the sodium ion insertion would also be the reason for the good stability of the GO and PGO membranes in nanofiltration. We have revised the related part of the manuscript. Supplementary Fig. 9,10 have been added.

5. In Figure 5d, the “permeance” should be “flux”.

Response/Action:

We thank the reviewer for his/her comments. We have revised the “permeance” by “flux”.

6. According to the AFM and TEM images, the pore size of PGO-9 is larger than that of PGO-6. But why

PGO-9 membrane has both lower permeance and salt rejection compared with PGO-6 membrane (Figure 5a,b,c)?

Response/Action:

We thank the reviewer for his/her comments. For GO membranes, the molecules have to pass through vertical defects/edges and shuttle back and forth in horizontal interlayer channels between adjacent nanosheets. The transport pathway is long and the tortuosity is extremely large. Because the PGO membranes had reduced transport pathway and tortuosity, the membrane showed increased performance than GO membrane. Moreover, the stacking property of GO membranes also affected the water permeance. As mentioned in the comments of reviewer 1, the excessively large holes are not to help permeation, because the holes will be filled by neighbouring nanosheets and the flexible GO nanosheets will curve to occupy the large empty space. For the PGO-9 membrane, the excessive membrane irregularity and nanosheet damage degree, which were identified by AFM, Raman, and XRD, induced the poorer water permeation than PGO-6.

For Na_2SO_4 desalination, the rejection was mainly affected by Donnan effect. Strong electrostatic repulsion of negatively charged surface towards SO_4^{2-} endowed the membranes with good rejection. Consequently, the maximum rejection was achieved by the PGO-3 membrane with largest electronegativity. It should be pointed out that the slightly smaller zeta potential of PGO-6 than GO at pH of ~7 seemed difficult to bring about substantial rejection amelioration. We calculated the permeation rates and water/salt selectivity of various membranes. We found that the Na_2SO_4 permeation rate of various membranes was roughly ordered by $\text{PGO-3} < \text{GO} < \text{PGO-6} < \text{PGO-9}$. The lowest Na_2SO_4 permeation rate of PGO-3 was attributed its largest electronegativity. The higher rejection of the PGO-6 membrane than GO was

explained by that the increment of water permeation rate was greater than that of Na_2SO_4 . The PGO-9 membrane had similar or higher Na_2SO_4 permeation rate than PGO-6. This was explained by that the more uniform stacking of the PGO-6 membrane could provide more uniform electronegative sub-nanometre interlayer than PGO-9 nanosheets with excessive membrane irregularity, thereby facilitating the repulsion for Na_2SO_4 . Based the difference in increment of water permeation rate and Na_2SO_4 permeation rate, the PGO-6 membrane had better performance than PGO-9 one.

We have added the discussion about the separation mechanism for the better performance of the PGO-6 membrane than PGO-9.

“Less permeance upgradation of PGO-3 than PGO-6 was explained by their inferior porosity; while the poorer permeance of PGO-9 than PGO-6 was attributed to the excessive membrane irregularity from nanosheet damage and the occupation of neighbouring nanosheets to large space of holes.”

“A similar or lower Na_2SO_4 permeation rate of PGO-6 than PGO-9 was attributed to that the more regular stacking provided more uniform electronegative sub-nanometre interlayer channels for better electrostatic repulsion.”

7. In Figure 5h, the separation performance of other previously reported porous or holey GO membranes should also be included and discussed.

Response/Action:

We thank the reviewer for his/her comments. We have added the Na_2SO_4 separation performance of other previously reported porous or holey GO membranes in Fig. 5h and Supplementary Table 4.

“Moreover, compared with recently reported high-performance GO-based membranes after chemical

modification and hole generation (Fig. 5h and Supplementary Table 4), the separation efficiency of the PGO membranes were still exceptional and competitive.”

Fig. 5. h. Comparison of the PGO membranes with commercial, GO, and porous GO membranes for Na_2SO_4 desalination. Desalination performance of most GO membranes is located in the dark grey region. Some recently reported GO membranes after modification with advanced separation performance are presented specially (Supplementary Table 4).

Supplementary Table 4. Separation performance of some recently reported GO membranes after modification with advanced performance. TBO: toluidine blue O; SA: sodium alginate; PAA: polyacrylic acid; PSS: polystyrene sulfonate; r-HGO: reduced holey GO membranes; and rNPGO: reduced nanoporous GO membrane.

Membrane	Solute	Concentration	Permeance	Rejection	Ref
----------	--------	---------------	-----------	-----------	-----

		(ppm)	($L m^{-2} h^{-1} bar^{-1}$)	(%)	
rGO	Na₂SO₄	2000	6	96	50
ZIF-8@GO	Na₂SO₄	100	49.8	52.9	25
NF-270	Na₂SO₄	100	10.1	91.5	25
GO	Na₂SO₄	100	2.1	84.3	25
GO	Na₂SO₄	14200	0.7	84	17
GO+9.1wt% TBO	Na₂SO₄	14200	0.4	90	17
GO-SA	Na₂SO₄	50	20.2	88.4	18
GO-PAA	Na₂SO₄	50	14.3	95.3	18
GO-PSS	Na₂SO₄	50	16.8	97.1	18
rHGO	Na₂SO₄	2000	9.2	96.7	49
rHGO	Na₂SO₄	2000	14.0	91.1	49
rNPGO	Na₂SO₄	2840	39.9	90.0	51

8. The applied voltage determined the pore size and chemical structure of PGO. But for a chemical reaction, the reaction time is also an important parameter. What is the electrochemical oxidation time or retention time of GO in the anode chamber? And how does the retention time affect the perforation of GO?

Response/Action:

We thank the reviewer for his/her suggestions. The nanowire electrochemical perforation of GO for holes formation depended on the reaction strength of direct and indirect oxidation and the reaction time. The applied voltages determined the nanowire anode reaction strength of direct and indirect oxidation, and the

flow rate was related to the reaction time of GO oxidation in the porous Co_3O_4 -NW anode. Both the stronger oxidation strength and longer reaction time can promote the GO perforation. Overall, the purpose of this work was to provide a facile, scalable, and controllable nanowire electrochemical concept for perforating and modifying nanosheets to adjust accessible characteristics. Hence, we evaluated the effect of applied voltages on the chemical structures and configurations of GO, and demonstrated the feasibility and superiority of nanowire electrochemical oxidation. We greatly appreciate the reviewer's valuable comments, and we will further improve the relevant work in the future.

9. Page 17, Line 1. The authors wrote: "the nanofiltration was mainly affected by Donnan effect." And as shown in Figure 4g, the Zeta potential of the GO and PGO membranes depends on pH. So the pH and salt concentrations will affect salt and dye rejections of PGO membranes. It's important to know the effect of pH and salt concentrations on separation performance in practical applications.

Response/Action:

We thank the reviewer for his/her comments. We have measured the separation performance of the PGO-6 membrane prepared with loading of 100 μg for Na_2SO_4 solution with different concentrations. Because of the less Donnan effect with the increase of salt concentration, the membrane showed slight decrease in rejection as the salt concentration increased. For the effect of pH, we greatly appreciate the reviewer's valuable comments and we will further preform the relevant work in the future.

We have added the discussion and Supplementary Fig. 22 about the separation performance of the PGO-6 membranes for Na_2SO_4 solution with different concentrations.

"We measured the NaCl desalination performance and investigated the effect of Na_2SO_4 concentration

on performance of the PGO-6 membrane. Low NaCl rejection of 35.2% and slightly decreased rejection for higher Na_2SO_4 concentration (Supplementary Fig. 22) suggested that the nanofiltration was mainly affected by Donnan effect.”

Supplementary Fig. 22. Water permeance and rejection of the PGO membrane prepared with loading of 100 μg for Na_2SO_4 solution with different concentrations.

Reviewer #3 (Remarks to the Author):

Summary

1. The approach mentioned in the following works is way simpler and straightforward than the presented approach: - Buelke, C., Alshami, A., Casler, J., Lin, Y., Hickner, M. and Aljundi, I.H., 2019. Evaluating graphene oxide and holey graphene oxide membrane performance for water purification. *Journal of Membrane Science*, 588, p.117195. - Lin, Y., Han, X., Campbell, C.J., Kim, J.W., Zhao, B., Luo, W., Dai, J., Hu, L. and Connell, J.W., 2015. Holey graphene nanomanufacturing: Structure, composition, and electrochemical properties. *Advanced Functional Materials*, 25(19), pp.2920-2927. is probably a better method for hole production: more controlled, smaller holes.

Response/Action:

We thank the reviewer for his/her comments. It is possible that we have not comprehensively discussed the characteristics of nanowire electrochemical method for perforating and modifying GO nanosheets, thereby making that the reviewer unable to clearly understand the simplicity and straightforwardness of the nanowire electrochemical method.

As the reviewer's comments, air oxidation has been used to generate holes in graphene by removing defect carbons in graphene. For obtaining porous GO nanosheets and separation membranes, the graphene with holes should be further oxidized by Hummers' method. The raw material for air oxidation is graphene powder. So far, the graphene is usually fabricated by reduction of GO, mechanical exfoliation of graphite, and chemical vapour deposition. The latter two methods are of low efficiency and have complex process. Moreover, the defect carbons in graphene from these two methods are few, which will limit the formation of holes in graphene by air oxidation. Based the information about raw graphene powder (Vor-X; grade:

reduced 070; lot: BK-77x) in these studies, it was possible that the graphene powder was reduced from GO. That was to say, for obtaining porous GO nanosheets by air oxidation, it might need the follow six steps: 1) oxidation of graphite by Hummers' method or other methods to fabricate graphite oxide, 2) exfoliation graphite oxide to obtain graphene oxide, 3) reduction of graphene oxide by thermal or chemical methods to prepare graphene powder, 4) air oxidation of graphene powder to form graphene powder with holes, 5) re-oxidation of graphene powder by Hummers' method or other methods, and 6) then re-exfoliation graphene oxide powder to obtain graphene oxide nanosheets. Comparatively, for nanowire electrochemical method, a more straightforward process with three steps was needed: 1) oxidation of graphite by Hummers' method or other methods to fabricate graphite oxide, 2) exfoliation graphite oxide to obtain graphene oxide, and 3) nanowire electrochemical method for obtaining porous GO nanosheets. In addition, relative to air oxidation with intermittent batch processing at high temperature, the nanowire electrochemical method with continuous flow-through process at room temperature might be benefit for the scalable, controllable, environment-friendly, mild, economical, and safe production of porous GO.

We have added the discussion and Supplementary Table 2 about the features of nanowire electrochemical method.

“So far, some other methods, e.g., chemical etching and air oxidation, have also been reported for generation of holes in GO nanosheets (Supplementary Table 2)⁴⁸⁻⁵³. Compared with hydrogen peroxide-needed chemical etching and high-temperature air oxidation, which had intermittent batch processing for several hours, the nanowire electrochemical strategy reported here, with continuous flow-through procedures, room temperature condition, and GO suspension flux over 1800 L m⁻² h, might have superiorities in operability, controllability, scalability, environmental friendliness, economy, safety, and efficiency. Moreover, relative to the previously reported methods, which might involve the reactions at

oxygen-containing regions of GO or at defect regions of graphene, the nanowire electrochemical method could perform oxidation and perforation for both graphitic and oxygen-containing regions because of the strong electrochemical reactions over nanowire tips.”

Supplementary Table 2. Comparison between nanowire electrochemical perforation and other methods.

RT: Room temperature; and NMP: N-methylpyrrolidone.

Method	Operation	Reagent	Condition	Raw	C/O	Ref.
Chemical etching	Batch	NH_4OH and H_2O_2	50 °C for 1–5 h	Liquid	Higher	48
Chemical etching	Batch	H_2O_2	100 °C for 1–4 h	Liquid	-	49
Chemical etching	Batch	$KMnO_4$, oxalic acid, and hydrochloric acid	RT for ~6 h	Liquid	Similar	50
Chemical etching	Batch	H_2O_2	70 °C for 10 h	Liquid	Similar	51
Air oxidation processing graphene and oxidation	Batch	Air	395–460 °C for 10 h	Solid	-	52,53
Thermal annealing and microwave treatment	Batch	Air and NMP	200 °C for 10 min	Solid	Higher	27
Nanowire electrochemical	Flow-through	$NaCl$	RT	Liquid	Lower or similar	This work

References

48. Wu, T., Moghadam, F. & Li, K. High-performance porous graphene oxide hollow fiber membranes with tailored pore sizes for water purification. *J. Membr. Sci.* **645**, 120216 (2022).
49. Chen, X. et al. Reduced holey graphene oxide membranes for desalination with improved water permeance. *ACS Appl. Mater. Interfaces* **12**, 1387-1394 (2020).
50. Ying, Y., Sun, L., Wang, Q., Fan, Z. & Peng, X. In-plane mesoporous graphene oxide nanosheet assembled membranes for molecular separation. *RSC Adv.* **4**, 21425-21428 (2014).
51. Li, Y. et al. Thermally reduced nanoporous graphene oxide membrane for desalination. *Environ. Sci. Technol.* **53**, 8314-8323 (2019).
52. Buelke, C. et al. Evaluating graphene oxide and holey graphene oxide membrane performance for water purification. *J. Membr. Sci.* **588**, 117195 (2019).
53. Lin, Y. et al. Holey graphene nanomanufacturing: structure, composition, and electrochemical properties. *Adv. Funct. Mater.* **25**, 2920-2927 (2015).
27. Kang, J. et al. Microwave-assisted design of nanoporous graphene membrane for ultrafast and switchable organic solvent nanofiltration. *Nat. Commun.* **14**, 901 (2023).

2. Concerns that all GO was being converted into PGO via the nanowires. What is more likely is that only some of the GO was actually reacted, and that some of those reactions produced PGCl since the electrolyte solution this was being done in consisted of 2.9 mg/mL NaCl.

Response/Action:

We thank the reviewer for his/her comments. For the possibility that only some of the GO was actually reacted mentioned by the reviewer, the simplified schematic illustration in Fig. 2a of nanowire perforation for GO nanosheets might make the reviewer confused that the reaction chamber was cylindrical. Actually, we used titanium net with pore sizes around 2.0 mm as cathode and selected porous graphite felt with fibre diameters of $\sim 10.0\ \mu\text{m}$ and pore sizes between the fibres of 50–200 μm as anode (Fig. 2a and Supplementary Fig. 2). Then the $\text{Co}_3\text{O}_4\text{-NW}$ was grown on the fibre for perforating GO nanosheets. For perforation, the GO suspension was continuously pumped into the electrochemical apparatus with flow rate of $15\ \text{mL min}^{-1}$ and under applied voltages of 3–9 V. The GO suspension flowed through the pores of cathode and anode sequentially. Therefore the GO nanosheets were modified uniformly. Moreover, as shown in the AFM images, we could observe that the GO nanosheets were not partially perforated.

For the production of PGCl mentioned by the reviewer, we have investigated the prepared GO and PGO membranes by using XPS. The results indicated that the GO, PGO-3, PGO-6, and PGO-9 had sodium and chlorine contents of 3.0–3.5% and 1.5–1.9%, respectively. The more sodium content than chlorine one for all membranes was attributed to the sodium insertion from the interaction between sodium and oxygen-containing groups. It should be noted that the PGO membranes showed similar chlorine content as GO, suggesting no formation of PGCl. Moreover, there was no C-Cl in the C 1s XPS profiles and the similar Cl XPS peak shape for all membranes also indicated that the PGO was not PGCl.

We have added the information about the apparatus and process in Fig. 1a for better understanding nanowire electrochemical perforation and added the Cl 2p and Na 1s XPS spectra of the GO, PGO-3, PGO-6, and PGO-9 nanosheets in Supplementary Fig. 9,10 for demonstrating the formation of PGO, rather than PGCl.

“The similar chlorine content, the nonexistence of C-Cl in C 1s XPS, and the similar Cl 2p peak shape implied the limited reaction between Cl⁻ and GO (Supplementary Fig. 9). The higher sodium content than chlorine one was explained by the stronger interaction of GO to Na⁺ (Supplementary Fig. 10).”

Fig. 2. a, Schematic illustration of nanowire perforation for GO nanosheets by using an electrochemical apparatus with a flow-through mode. GO nanosheets in suspension flowed through the cathode and anode sequentially, and were transformed to porous nanosheets at Co₃O₄-NW anode with applied voltages of 3–9 V. **b,c**, SEM images of Co₃O₄-NW and Co₃O₄-NS on fibre of porous graphite felt (SEM image in a).

Supplementary Fig. 9. Cl 2p XPS spectra of the GO, PGO-3, PGO-6, and PGO-9 nanosheets. It should be noted that the PGO membranes showed similar chlorine content as GO, suggesting no formation of PGCl. Moreover, no C-Cl in the C 1s XPS and the similar Cl XPS peak shape for all membranes also indicated that the PGO was not PGOCl.

Supplementary Fig. 10. Na 1s XPS spectra of the GO, PGO-3, PGO-6, and PGO-9 nanosheets. The GO, PGO-3, PGO-6, and PGO-9 had sodium and chlorine contents of 3.0–3.5% and 1.5–1.9%, respectively. The more sodium content than chlorine one for all membranes was attributed to the sodium insertion from the interaction between sodium ion and oxygen-containing groups. The sodium ion insertion would be beneficial to the stability of the GO and PGO membranes in nanofiltration¹⁵.

3. Linear scanning voltammetry is not a good method for measuring PGO production in this setup.

Response/Action:

We thank the reviewer for his/her comments. In this study, linear scanning voltammetry was used to validate the existence of direct GO electrochemical reaction, rather than used to measure PGO production.

Please see more details in the Response/Action of Method comment 4.

4. Inaccurate claims about XPS data

Response/Action:

We thank the reviewer for his/her comments. Please see more details in the Response/Action of Characterization comment 2.

5. Low Na₂SO₄ concentrations used, and no data shown for NaCl tests besides < 40% rejection

Response/Action:

We thank the reviewer for his/her comments. The PGO-3 and PGO-6 membranes showed NaCl rejection of 39.2% and 35.2%, respectively. We have added the rejection data in manuscript.

6. No control (bare PES) membrane test! Strongly suggested that they do this

Response/Action:

We thank the reviewer for his/her comments. The PES substrate was microfiltration membrane with pore size of 0.22 μm and had no rejection for salts. Please see more details in the Response/Action of Performance comment 2.

7. Full read for grammar

Response/Action:

We thank the reviewer for his/her comments. We have improved the language of manuscript.

Abstract

1. “It remains great challenging to design nanosheets with uniform interior nanopores.” Lin 2015 showed that open air tube furnace heating at high temps over a short or long period of time produced holey graphene with small or large, but uniform, holes.

Response/Action:

We thank the reviewer for his/her comments. We have rewritten abstract based the comments of reviewer 1 and reviewer 3.

“Two-dimensional nanosheets, e.g., graphene oxide (GO), have been widely used to fabricate efficient membranes for molecular separation. However, because of poor transport across nanosheets and high width-to-thickness ratio, the permeation pathway length and tortuosity of these membranes are extremely large, which limit their separation performance. Here we report a facile, scalable, and controllable nanowire electrochemical concept for perforating and modifying nanosheets to shorten permeation pathway and adjust transport property. It is found that confinement effects with locally enhanced charge density, electric field, and hydroxyl radical generation over nanowire tips on anode can be executed under low voltage, thereby inducing confined direct electron loss and indirect oxidation to reform configuration

and composition of GO nanosheets. We demonstrate that the porous GO nanosheets with a lot of holes are suitable for assembling separation membranes with tuned accessibility, tortuosity, interlayer space, electronegativity, and hydrophilicity. For molecular separation, the prepared membranes exhibit quadruple water permeance and higher rejections for salts (> 91%) and small molecules (> 96%) as/than original ones. This nanowire electrochemical perforation concept offers a feasible strategy to reconstruct two-dimensional materials and tune their transport property for separation.”

Method

1. Did not list ultrasonication time for exfoliating GO in water, although we can assume that it was well suspended. Also centrifugation/decantation to remove unexfoliated material, but didn't list duration at 4000 RPM.

Response/Action:

We thank the reviewer for his/her comments. The duration was 5 min. We have added the details in the section of Preparation of GO suspension.

2. For the nanowire perforation of the GO sheets, I have serious concerns about the yield of perforated GO (PGO). The nanowire electrode apparatus has a diameter of 5 mm, with flow velocity of 12.5 mm/s. So relative to the reaction chamber, the fluid is moving at a decent pace. However, the nanowires themselves are only 3 microns long along the edge of the chamber, so 99.94% of the reaction chamber is flowing bulk fluid. While the anode will attract GO to the nanowires via electrophoresis, I wonder how much actual PGO is formed. Assuming the perforation is instantaneous, the question becomes “how much GO migrates

to the nanowires and then leaves per second?” My guess is, while the majority of GO is indeed perforated, there remains quite a bit that exits the chamber unreacted.

Response/Action:

We thank the reviewer for his/her comments. There may have misunderstandings about experimental device and nanowire perforation. As mentioned in the response of the Summary comment 2, for perforation, the GO suspension flowed through the pores of cathode and anode sequentially. The porous graphite felt with fibre diameters of $\sim 10.0\ \mu\text{m}$ and pore sizes between the fibres of $50\text{--}200\ \mu\text{m}$ was used as anode, and the $\text{Co}_3\text{O}_4\text{-NW}$ was grown on the fibre for perforating GO nanosheets. Therefore, the GO nanosheets were modified uniformly. Moreover, as shown in the AFM images, we could observe that the GO nanosheets were not partially perforated. We have revised the Fig. 2a by adding the pictures of Ti net cathode and $\text{Co}_3\text{O}_4\text{-NW}$ anode for better understanding. We also evaluated the PGO concentration after treatment by the electrochemical apparatus via UV-Vis spectroscopy (*Science* **342**, 95-98 (2013)). The similar concentration of the PGO suspensions after perforation as original GO suspension indicated that the GO nanosheets had not been intercepted by the porous electrodes. The nanowire perforation with a flow-through mode could achieve continuous PGO production.

3. Additionally, the concentration of GO used in this process is $0.02\ \text{mg/mL}$. However, it was not done in DI water, and instead they used NaCl as the electrolyte solution during the perforation process. The NaCl concentration was $2.9\ \text{mg/mL}$, so I also have to wonder now that, not only “how much GO exited the chamber unreacted”, but “how much GO was instead converted to graphene chloride (GCl)?” They make the case that electrolyzed water from the cathode can form OH^- , which can then attach to the nanowire tips

at the anode to reduce to OH-dot, so they are aware that other side reactions are occurring and material other than GO are migrating to the nanowires.

Response/Action:

We thank the reviewer for his/her comments. The deionized (DI) water has resistance $\geq 18.2 \text{ M}\Omega/\text{cm}$ with negligible conductivity, which will increase the ohmic drop between the electrodes for inhibiting the electrochemical reactions. Hence, we used 50 mM NaCl with conductivity of $4700 \text{ }\mu\text{S}/\text{cm}$ as electrolyte to induce the electrochemical reactions at low applied voltages.

For “how much GO was instead converted to graphene chloride (GCl)”, as mentioned in response of Summary comment 2, the similar chlorine content and Cl XPS peak for the GO and PGO membranes and no C-Cl in the C 1s XPS profiles of PGO membranes suggested almost no formation of PGCl.

4. Along those same lines, they measure the reaction occurrence at the nanowire tip via linear scanning voltammetry, and they equate this to PGO production: “there’s a change in voltage, so PGO is being produced.” However, since there is much more activity at the nanowire tip other than the PGO reaction that would change the voltage, it’s not a reliable metric for PGO production.

Response/Action:

We thank the reviewer for his/her comments. We have not equated the reaction occurrence at the nanowire tip via linear scanning voltammetry to PGO production. The formation of PGO nanosheets was confirmed by various characterizations. Linear scanning voltammetry (LSV) was used to explore the occurrence of direct and indirect oxidation via using the solution with or without GO addition. The electrochemical

properties of the electrodes were measured with a three-electrode system using 3.5 M Ag/AgCl as reference electrode. The applied voltage (V) is totally different from the electrode potential (V vs. 3.5 M Ag/AgCl). The electrode potentials (V vs. 3.5 M Ag/AgCl) were recorded under applied voltages from 2.0 to 9.0 V (Fig. 2e). Linear scanning voltammetry was used to record the current (strength of electrochemical redox) of the Co₃O₄-NW anode at different anode potentials (V vs. 3.5 M Ag/AgCl), and the polarization potential means that electrochemical oxidation on Co₃O₄-NW anode begin to occur with significant increase in current. Hence, without GO addition in influent, the electrochemical oxidation on Co₃O₄-NW anode were attributed to the OH⁻ ions oxidation to •OH radicals. While, with GO addition in influent, the electrochemical reaction on anode were attributed to the oxidation reactions of OH⁻ ions GO nanosheets. Based on the above basic electrochemical theory, the lower polarization potential (~0.91 V vs. Ag/AgCl) for the GO suspension than that for the solution without GO (~1.01 V vs. Ag/AgCl) validated the existence of direct GO electrochemical reaction (Fig. 2f). For the solution without GO, the Co₃O₄-NW anode displayed lower potential (~1.01 V vs. Ag/AgCl) for OH⁻ electrolysis occurrence than Co₃O₄-NS (~1.15 V vs. Ag/AgCl) as well.

Characterization

1. For AFM graphs, they state that GO was monolayer at ~1 nm, and that PGO was thicker at around 1.8 nm. They suggest that this increase in thickness is due to “grafted out-of-plane functional groups”, but the XPS data shows little change in functional groups. It most likely instead is a bilayer of PGO. Perhaps the perforation reaction can cause some GO sheets to crosslink

Response/Action:

We thank the reviewer for his/her comments. We have measured the thickness of different nanosheets. It was almost impossible that all measured nanosheets were bilayer and the two nanosheets were overlapped completely. Therefore, we did think that the grafted out-of-plane oxygen-containing groups induced the increased in thickness. As well, the more water absorption of PGO from their better hydrophilicity might also contribute to the thickness increase.

2. Speaking of XPS data, they state that the C 1s peak areas were expanded, and that the COOH and C=O peaks for the O 1s graphs were more intense. Neither show much change for PGO6 and PGO9, with some change for PGO3. from what I can tell. Figures 3n and 3o also confirm this?

Response/Action:

We thank the reviewer for his/her comments. In fact, we stated “the peak areas of oxygen-containing groups in the C 1s profiles of PGO were expanded, and the PGO-3 nanosheets had highest C-OH/C-O-C/C=O/COOH content of 51.2%”, rather than the C 1s peak areas were expanded. For the O 1s spectra, the C=O and COOH peaks of PGO were more intensive than those of GO. Compared with the obvious increase of C-OH/C-O-C, C=O, and COOH for PGO-3 to GO, the PGO-6 and PGO-9 showed lower increase in C-OH/C-O-C, C=O, and COOH contents as shown in Fig. 3n,o and Supplementary Table 1. For nanowire perforation with voltage of 3 V, hydroxyl and epoxy decoration, carbon-carbon fracture and further oxidation to form carbonyl and carboxyl were dominant, thereby leading to the enrichment of oxygen-containing groups. For perforation at 6 V or 9 V, stronger direct and indirect oxidations led to oxidation and perforation for both graphitic and oxygen-containing regions, the oxygen-containing groups of PGO-6 and PGO-9 varied slightly as compared with GO.

3. For XRD, they don't list the PGO9 wet state d-spacing. However, from the graph, we can tell it would be more than 1.38 nm. Large increases seen here, even for their GO at 0.91 nm to 1.28 nm (we had an increase of 0.71 nm to 0.85 nm). Since they used such low loading on the membrane during vacuum filtration, my guess is that what little GO or PGO was there had an easier time to expand since there wasn't additional GO to keep it from expanding: sort of like "GO compressing GO", or self-compression.

Response/Action:

We thank the reviewer for his/her comments. In fact, many studies have reported the interlayer space increased from 0.80~0.95 nm to 1.2~1.5 nm by wetting, e.g., from 0.81 to ~1.38 nm (*Nat. Sustain.* **4**, 402-408 (2021)), from 0.9 to 1.4 nm (*Nat. Nanotechnol.* **12**, 1083-1088 (2017)), and from 0.9±0.1 nm to 1.3 nm (*Science* **343**, 752-754 (2014)). We are not sure the data sources of increase from 0.71 to 0.85 nm, possibly from the *J. Membr. Sci.* **588**, 117195 (2019). The difference in the increase of interlayer space may be attributed to the oxidation degree and wetting conditions.

Performance

1. Testing parameters are 500 mL/min at 2 bar for the cross-flow cell, 6 hours of compression before data collection, and concentrations of 50 mg/L for methyl blue, 0.5 mg/mL for Na₂SO₄, and unknown concentration for NaCl.

Response/Action:

We thank the reviewer for his/her comments. The concentration for NaCl was 500 mg L⁻¹. We have added

the details in the section of Separation performance evaluation.

2. They never did a control test for seeing PES membrane performance. This is critical! The large d-spacings when wet, combined with the low loading, leads me to believe the support membrane is doing a lot of the heavy lifting, and a control test would prove otherwise.

Response/Action:

We thank the reviewer for his/her comments. The PES substrate was microfiltration membrane with pore size of 0.22 μm . We had measured the Na_2SO_4 rejection of the PES substrate. The result indicated that almost no rejection could be achieved.

3. Supp Figure 17 shows PGO3 had best separation factor of water/ Na_2SO_4 across all loadings. PGO6 had highest permeance. This makes sense since PGO3 had no holes and slightly increased oxygen-containing functional groups, and PGO6 had holes, which were smaller than PGO9. Small holes are better than big holes? Yet another reason that open air tube furnace processing is better in my opinion: the entire sample undergoes controlled oxidation to form small holes.

Response/Action:

We thank the reviewer for his/her comments. For GO membranes, the molecules have to pass through vertical defects/edges and shuttle back and forth in horizontal interlayer channels between adjacent nanosheets. The transport pathway is long and the tortuosity is extremely large. Because the PGO membranes had reduced transport pathway and tortuosity, the membrane showed increased performance

than GO membrane. Moreover, the stacking property of GO membranes also affected the water permeance. As mentioned in the comments of reviewer 1, the excessively large holes are not to help permeation, because the holes will be filled by neighbouring nanosheets and the flexible GO nanosheets will curve to occupy the large empty space. For the PGO-9 membrane, the excessive membrane irregularity and nanosheet damage degree, which were identified by AFM, Raman, and XRD, induced the poorer water permeation than PGO-6.

For Na_2SO_4 desalination, the rejection was mainly affected by Donnan effect. Strong electrostatic repulsion of negatively charged surface towards SO_4^{2-} endowed the membranes with good rejection. Consequently, the maximum rejection was achieved by the PGO-3 membrane with largest electronegativity. It should be pointed out that the slightly smaller zeta potential of PGO-6 than GO at pH of ~ 7 seemed difficult to bring about substantial rejection amelioration. We calculated the permeation rates and water/salt selectivity of various membranes. We found that the Na_2SO_4 permeation rate of various membranes was roughly ordered by $\text{PGO-3} < \text{GO} < \text{PGO-6} < \text{PGO-9}$. The lowest Na_2SO_4 permeation rate of PGO-3 was attributed its largest electronegativity. The higher rejection of the PGO-6 membrane than GO was explained by that the increment of water permeation rate was greater than that of Na_2SO_4 . The PGO-9 membrane had similar or higher Na_2SO_4 permeation rate than PGO-6. This was explained by that the more uniform stacking of the PGO-6 membrane could provide more uniform electronegative sub-nanometre interlayer than PGO-9 nanosheets with excessive membrane irregularity, thereby facilitating the repulsion for Na_2SO_4 . Based the difference in increment of water permeation rate and Na_2SO_4 permeation rate, the PGO-6 membrane had better performance than PGO-9 one.

We have added the discussion about the separation mechanism for the better performance of the PGO-6 membrane than PGO-9.

“Less permeance upgradation of PGO-3 than PGO-6 was explained by their inferior porosity; while the poorer permeance of PGO-9 than PGO-6 was attributed to the excessive membrane irregularity from nanosheet damage and the occupation of neighbouring nanosheets to large space of holes.”

“A similar or lower Na_2SO_4 permeation rate of PGO-6 than PGO-9 was attributed to that the more regular stacking provided more uniform electronegative sub-nanometre interlayer channels for better electrostatic repulsion.”

4. Low rejection of NaCl at < 40%, and no other data presented. Since PGO3 had the best rejection of Na_2SO_4 , we can assume 40% rejection of NaCl was achieved with that membrane.

Response/Action:

We thank the reviewer for his/her comments. The PGO-3 and PGO-6 membranes showed NaCl rejection of 39.2% and 35.2%, respectively. We have added the rejection data in manuscript.

REVIEWER COMMENTS

Reviewer #1 (Remarks to the Author):

I find that authors answered my questions satisfactory and introduced revisions which significantly improved paper. I can still disagree on some points but this should not hinder publication of study. Abstract is better written now, hole size distribution added, tortuosity change due to holes analyzed. One advise which I can still give to authors is to add some citations about sorption of dyes by GO. It is not that rejection of dyes is solely explained by sorption. I did not mean this in my question. But dyes modify structure of GO significantly. Sorption of dye molecules change the properties of GO membranes including inter-layer distance. The filtration is then performed for membrane in dye-saturated state which is chemically different and size of "permeation channels" is modified. It is impossible to ignore sorption of MB when it adds 20-30% to the weight of membrane.

I checked also comments by other reviewers and authors reply. I think that adding review for other methods of making holey GO is sufficient to satisfy their most important concerns. While other methods to make holes are certainly available and have been reported previously, new method is of interest for broad audience.

Reviewer #2 (Remarks to the Author):

My previous concerns (1, 3, 8, and 9) were not addressed properly in the revisions. The advantages of the nanowire electrochemical method for perforating GO nanosheets were exaggerated in the revised manuscript. I still think the novelty and quality of this manuscript do not meet the high standard of Nature Communications.

Reviewer #3 (Remarks to the Author):

Although I was hesitant in recommending acceptance of this manuscript in Nature Comm at the onset, my hesitation is now gone after the revision. The authors' responses are sound and quite convincing. I believe they have addressed all my concerns satisfactorily, aside of my low enthusiasm on the overall membrane performance and salts rejection rates.

All in all, I am satisfied with the provided responses and ready to proceed with acceptance without further modifications.

**Response letter for “Nanowire electrochemical perforation of graphene oxide
nanosheets for membrane separation (NCOMMS-23-35238A-Z)”**

Reviewer #1 (Remarks to the Author):

I find that authors answered my questions satisfactory and introduced revisions which significantly improved paper. I can still disagree on some points but this should not hinder publication of study. Abstract is better written now, hole size distribution added, tortuosity change due to holes analyzed.

Response/Action:

We thank the reviewer for his/her constructive suggestions. We have further revised the manuscript to address the concerns of the reviewer.

One advise which I can still give to authors is to add some citations about sorption of dyes by GO. It is not that rejection of dyes is solely explained by sorption. I did not mean this in my question. But dyes modify structure of GO significantly. Sorption of dye molecules change the properties of GO membranes including inter-layer distance. The filtration is then performed for membrane in dye-saturated state which is chemically different and size of "permeation channels" is modified. It is impossible to ignore sorption of MB when it adds 20-30% to the weight of membrane.

Response/Action:

We thank the reviewer for his/her comments. We agree with the reviewer's comments that the dye sorption can modify the structures and interlayer channels of GO membranes. We have revised the manuscript as

reviewer's comments. The sentences of "As reported in previous studies^{44,59}, the adsorption of membranes for antibiotics and dyes occurred during filtration. Certainly, this adsorption could change the structure, interlayer space, and chemical composition of membranes, thereby affecting and contributing to separation." have been added. The related references have been added and discussed.

44. Boulanger, N. et al. Enhanced sorption of radionuclides by defect-rich graphene oxide. *ACS Appl. Mater. Interfaces* **12**, 45122-45135 (2020).

59. Nordenström, A. et al. Intercalation of dyes in graphene oxide thin films and membranes. *J. Phys. Chem. C* **125**, 6877-6885 (2021).

I checked also comments by other reviewers and authors reply. I think that adding review for other methods of making holey GO is sufficient to satisfy their most important concerns. While other methods to make holes are certainly available and have been reported previously, new method is of interest for broad audience.

Response/Action:

We thank the reviewer for his/her positive comments.

Reviewer #2 (Remarks to the Author):

My previous concerns (1, 3, 8, and 9) were not addressed properly in the revisions. The advantages of the nanowire electrochemical method for perforating GO nanosheets were exaggerated in the revised manuscript. I still think the novelty and quality of this manuscript do not meet the high standard of Nature

Communications.

Response/Action:

We thank the reviewer for his/her comments. We have properly addressed the reviewer's concerns in this revised version of the manuscript.

The previous concern 1 was that *“The existence of GO will change the pH of the solution, leading to the polarization potential difference of OH⁻ electrochemical reaction. I think these LSV tests can't prove the existence of direct GO electrochemical reaction.”* In previous revision, we have revised the manuscript as below. *“Theoretically, the acidic substances of GO nanosheets would neutralize the OH⁻ of cathode-treated influent and then increase the electrode potential for OH⁻ oxidation. However, the lower polarization potential (~0.91 V vs. Ag/AgCl) of Co₃O₄-NW for the GO suspension than that for the solution without GO (~1.01 V vs. Ag/AgCl) was observed. This phenomenon validated the existence of direct GO electrochemical reaction (Fig. 2f).”* In this version, we further measured the pH of the cathode-treated influent without GO addition. The results indicated that the pH of the cathode-treated influent without GO was higher than that of the cathode-treated influent with GO, confirming that the acidic substances from GO neutralized the OH⁻ ions in cathode-treated influent. Combining that the lower OH⁻ concentration theoretically required higher anode potential for OH⁻ oxidation to •OH and the experimental potential of the Co₃O₄-NW anode for the GO suspension was lower than that for the solution without GO, it was confirmed the existence of direct GO electrochemical reaction. We have further revised the related part of the manuscript and the pH of the cathode-treated influent without GO have been measured and added in Fig. 2d. The related discussion has been revised to *“Theoretically, the acidic substances of GO nanosheets would neutralize the OH⁻ of cathode-treated influent and then increase the electrode potential for OH⁻*

oxidation. As expected, the experimental results confirmed that the GO addition reduced the pH and the OH^- concentration of cathode-treated influent (Fig. 2d). However, the polarization potential (~ 0.91 V vs. Ag/AgCl) of Co_3O_4 -NW for the GO suspension was lower than that for the solution without GO (~ 1.01 V vs. Ag/AgCl). This phenomenon validated the existence of direct GO electrochemical reaction (Fig. 2f).”

Fig. 2. d, Solution pH of influent, influent after treatment with Ti net cathode, and effluent after treatment with Co_3O_4 -NW anode.

The previous concern 3 was that “In Figure 4e, it’s hard to identify the XRD peaks. They should be retested to get reliable interlayer spaces.” For the thin GO membranes, the XRD peak intensity is usually weak. For obtaining the XRD data with higher peak intensity, we have collected the XRD data of the thick GO membranes under slower scanning mode. As shown in Supplementary Fig. 17, the XRD peak position of the thick membranes was similar to that of the thin ones. The thick PGO membranes also showed XRD peak shift compared with GO, suggesting the expanded interlayer space. Unlike the thin PGO-9 membrane with almost no XRD peak, the thick PGO-9 membrane still had peak in the XRD pattern. However, as the

large holes and dilapidated configuration induced irregular nanosheet arrangement, the peak intensity of the thick PGO-9 membrane was much smaller than that of other thick membranes. These results confirmed the validity of the discussion based on the XRD data of the thin membranes and agreed with the Raman results. We have further revised the manuscript as reviewer's comment and added Supplementary Fig. 17 to show the XRD data with more intensive peak.

Supplementary Fig. 17. XRD patterns of the GO, PGO-3, PGO-6, and PGO-9 membranes with loading of 1600 μg . The XRD patterns (also shown in Fig. 4e) of the thin GO and PGO membranes with loading of 200 μg are presented for comparison. The XRD peak position of the thick membranes was similar to those of the thin membranes. After electrochemical perforation, the XRD peak of PGO shifted to lower degree compared with that of GO, suggesting the expanded interlayer space. Although the thick PGO-9 membrane

with characteristic peak was different from the thin PGO-9 membrane with almost no XRD peak, the peak intensity of the thick PGO-9 membrane was much smaller than that of other thick membranes due to the irregular nanosheet arrangement.

The previous concern 8 was that *“The applied voltage determined the pore size and chemical structure of PGO. But for a chemical reaction, the reaction time is also an important parameter. What is the electrochemical oxidation time or retention time of GO in the anode chamber? And how does the retention time affect the perforation of GO?”* We agree with the reviewer’s comment. The nanowire electrochemical perforation of GO for hole formation depended on the reaction strength and the reaction time. The applied voltage was related to reaction strength, while the flow rate was related to the retention time and reaction time. Theoretically, the stronger reaction strength and longer reaction time could induce the stronger perforation. In previous manuscript, we have demonstrated that the higher applied voltage could induce the formation of larger holes in the PGO nanosheets. For reaction time, based on the flow rate of 15 mL min^{-1} , the retention time of the GO suspension in the porous anode was calculated as 9.8 s. In order to investigate the effect of reaction time on perforation, the flow rate of the GO suspension was increased to 30 mL min^{-1} . Correspondingly, the retention times was reduced to 4.9 s. Compared with the PGO-6 nanosheets perforated with retention time of 9.8 s, the PGO-6 perforated with retention time of 4.9 s showed smaller holes due to the reduced contact time and reaction time. We have revised the manuscript. The sentences of *“Nanowire electrochemical perforation of GO depended on reaction strength and reaction time. Besides perforation at various applied voltages for GO suspension with flow rate of 15 mL min^{-1} and retention time of 9.8 s, nanowire perforation was further performed with flow rate of 30 mL min^{-1} and retention time of 4.9 s. As expected, the PGO-6 nanosheets with shorter retention time showed smaller*

holes (Supplementary Fig. 8). These results demonstrated the controllability of nanowire electrochemical perforation.” The AFM images of the PGO-6 nanosheets perforated with retention time of 4.9 s have been added in Supplementary Fig. 8.

Supplementary Fig. 8. AFM images of the PGO-6 nanosheets perforated with flow rate of 30 mL min^{-1} and retention time of 4.9 s.

The previous concern 9 was that “Page 17, Line 1. The authors wrote: “the nanofiltration was mainly affected by Donnan effect.” And as shown in Figure 4g, the Zeta potential of the GO and PGO membranes depends on pH. So the pH and salt concentrations will affect salt and dye rejections of PGO membranes. It’s important to know the effect of pH and salt concentrations on separation performance in practical applications.” In previous revision, we have measured the separation performance of the PGO-6 membrane prepared with loading of $100 \mu\text{g}$ for Na_2SO_4 solution with different concentrations. Because of the less Donnan effect with the increase of salt concentration, the membrane showed slight decrease in rejection as the salt concentration increased. In this revision, we have investigated the effect of pH on performance of the PGO-6 membrane as the reviewer’s comments. As shown in Supplementary Fig. 25,

the Na_2SO_4 rejection decreased at pH of 4 due to the reduced Donnan effect. However, for antibiotic removal, the tetracycline separation efficiency was kept at high level under different pH due to the size exclusion separation mechanism. We have presented the separation performance of the PGO-6 membrane for salt and antibiotic solutions at different pH in Supplementary Fig. 25. “We measured the NaCl desalination performance and investigated the effect of Na_2SO_4 concentration on performance of the PGO-6 membrane. Low NaCl rejection of 35.2% and slightly decreased rejection for higher Na_2SO_4 concentration suggested that the nanofiltration was mainly affected by Donnan effect⁵¹” has been revised to “We measured the NaCl desalination performance and investigated the effects of Na_2SO_4 concentration and solution pH on performance of the PGO-6 membrane. Low NaCl rejection of 35.2% and decreased rejection for Na_2SO_4 solution with higher concentration and lower pH suggested that the nanofiltration was mainly affected by Donnan effect⁵¹ (Supplementary Fig. 24,25)”

Supplementary Fig. 25. Rejection of the PGO-6 membrane prepared with loading of 100 μg for Na_2SO_4 and TET (tetracycline) solutions with different pH at pressure 2 bar. The Na_2SO_4 rejection decreased at pH

of 4 due to the reduction of Donnan effect. However, for tetracycline removal, the separation efficiency was kept at high level under different pH due to the size exclusion separation mechanism.

Reviewer #3 (Remarks to the Author):

Although I was hesitant in recommending acceptance of this manuscript in Nature Comm at the onset, my hesitation is now gone after the revision. The authors' responses are sound and quite convincing. I believe they have addressed all my concerns satisfactorily, aside of my low enthusiasm on the overall membrane performance and salts rejection rates. All in all, I am satisfied with the provided responses and ready to proceed with acceptance without further modifications.

Response/Action:

We thank the reviewer for his/her comments and constructive suggestions.

REVIEWERS' COMMENTS

Reviewer #1 (Remarks to the Author):

I think that authors responded to reviewers questions in adequate way. I recommend to accept this paper for publication.

Reviewer #2 (Remarks to the Author):

Although I am still suspicious of the practicality of the nanowire electrochemical method for perforating GO nanosheets, the responses of the authors have addressed most of my concerns. I think the revised manuscript is sound and ready to be accepted without further modifications.

**Response letter for “Nanowire electrochemical perforation of graphene oxide
nanosheets for membrane separation (NCOMMS-23-35238-B)”**

Reviewer #1 (Remarks to the Author):

I think that authors responded to reviewers questions in adequate way. I recommend to accept this paper for publication.

Response/Action:

We thank the reviewer for his/her positive comments.

Reviewer #2 (Remarks to the Author):

Although I am still suspicious of the practicality of the nanowire electrochemical method for perforating GO nanosheets, the responses of the authors have addressed most of my concerns. I think the revised manuscript is sound and ready to be accepted without further modifications.

Response/Action:

We thank the reviewer for his/her positive comments.